# Future-aware Safe Active Learning of Time Varying Systems using Gaussian Processes

**Markus Lange-Hegermann**                                          *markus.lange-hegermann@th-owl.de*
*Department of Electrical Engineering and Institute Industrial IT - inIT*
*OWL University of Applied Sciences and Arts*
*Lemgo, Germany*

**Christoph Zimmer**                                          *Christoph.Zimmer@de.bosch.com*
*Bosch Center for Artificial Intelligence*
*Renningen, Germany*

**Reviewed on OpenReview:** *https: // openreview. net/ forum? id= YBPbMKJbLd*

## Abstract

Experimental exploration of high-cost systems with safety constraints, common in engineering applications, is a challenging endeavor. Data-driven models offer a promising solution, but acquiring the requisite data remains expensive and is potentially unsafe. Safe active learning techniques prove essential, enabling the learning of high-quality models with minimal expensive data points and high safety. This paper introduces a safe active learning framework tailored for time-varying systems, addressing drift, seasonal changes, and complexities due to dynamic behavior. The proposed Time-aware Integrated Mean Squared Prediction Error (T-IMSPE) method minimizes posterior variance over current and future states, optimizing information gathering also in the time domain. Empirical results highlight T-IMSPE's advantages in model quality through synthetic and real-world examples. State of the art Gaussian processes are compatible with T-IMSPE. Our theoretical contributions include a clear delineation which Gaussian process kernels, domains, and weighting measures are suitable for T-IMSPE and even beyond for its non-time aware predecessor IMSPE.

## 1 Introduction

Many systems pose challenges due to their high costs for experimentation, whether in actual implementation or in numerical simulation of the underlying physics. This concern is particularly pronounced in engineering applications. Data-driven models emerged as a valuable solution for swiftly simulating such systems. However, acquiring the necessary data for these models can still be prohibitively expensive, e.g. in machine calibration (Badra et al., 2020), biology (Sverchkov & Craven, 2017), numerical simulation (Al-Obaidi, 2022), prototype building (Torres et al., 2016), or process optimization (Hernández Rodríguez et al., 2022). To address this situation where measurements can be taken at almost arbitrary positions in a continuous domain, *active learning* techniques become crucial, allowing the learning of a high-quality model using a minimal amount of expensive data points. This involves setting inputs or calibration parameters in a manner that maximizes the information gathered from experiments.

Moreover, many real world systems are subject to *safety constraints*, such as critical temperatures, pressures, or restricted sensor working conditions, which must be adhered to during measurements. Often, these safety constraints are unknown and need to be modeled, leading to the application of safe active learning and related methodologies (Berkenkamp et al., 2023; Cowen-Rivers et al., 2022; Sui et al., 2018; Schreiter et al., 2015). While active learning is usually superior to static Design of Experiments (DoE) approaches (Smith, 1918; Pukelsheim, 2006), static DoE is not even applicable in the face of learnable safety constraints on outputs.

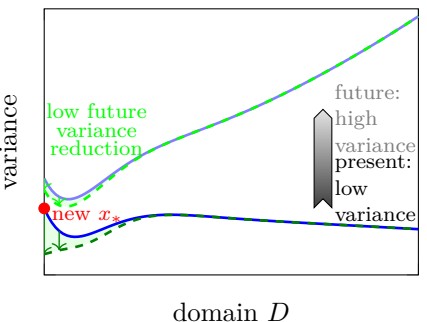 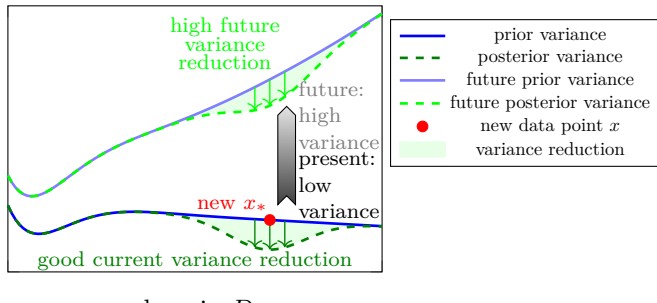

(a) Entropy, measure high current variance    (b) T-IMSPE (ours), reduce future variance

Figure 1: This illustrative sketch showcases the efficacy of our acquisition function T-IMSPE in variance reduction on a domain $D$ in present (dark color) and future (light color). The darker color at the bottom exhibits the transition from the prior blue line to the posterior green dashed line when conditioning on the red data point $x$ at present time. The lighter colored top delineates the induced variance reduction in a future time step. The left subplot (a) highlights the suboptimal variance reduction when employing the entropy acquisition function: it leads to a measurement at the boundary and to a very low reduction of the future variance. The right subplot (b) reveals the capability of T-IMSPE to outperform entropy in reducing the average variance at current and future time steps.

Real-world scenarios frequently involve *time-varying systems*, presenting additional complexities. Examples include systems affected by drift (Talluru et al., 2014), possibly caused by continuously polluting sensors (Thewes et al., 2015); systems influenced by periodic trends like seasonal variations (Bacastow et al., 1985; Scherliess & Fejer, 1999); or dynamic systems, where current decisions influence future behavior, such as is reflected in NX (Non-linear eXogenous) model structures (Chen & Huang, 2015; Zimmer et al., 2018); Actively learning such systems proves challenging, as parts of the system may change or certain operation points might be currently unreachable. Despite these challenges, there is a substantial demand for learning such systems. For example production systems with periodic changes may require accurate model-based anomaly detection, drift in sensors needs to be counteracted, or engines needs to be modeled for a cost-efficient calibration. Dynamic systems necessitate an understanding of their dynamic behavior.

This paper introduces a safe active learning framework tailored for time-varying systems. The approach involves measuring at safe positions where the maximum information can be collected. Notably, we focus on also *gathering information about future scenarios*, rather than solely the current system state.

Technically, we start from the acquisition function IMSPE (Integrated Mean Squared Prediction Error), also known under various other names such as ALC (Active Learning Cohn) or IMSE (Integrated Mean Squared Error) (Sacks et al., 1989b). This acquisition function chooses measurement points that minimize the posterior prediction variances, integrated over the entire relevant domain. We extend IMSPE to the acquisition function Time-aware Integrated Mean Squared Prediction Error (T-IMSPE), which minimizes the posterior variance over *current and future states* of the systems. Surprisingly, the necessary integration in both IMSPE and T-IMSPE can be computed in closed form for many Gaussian process models, in particular the ones that are state of the art in active learning scenarios.

Our main contributions are:

- We introduce the novel T-IMSPE acquisition function for safe active learning of time-varying systems, see Figure 1. We compute the defining integral in closed form, improving computational complexity and accuracy.

- We demonstrate the advantages of T-IMSPE in terms of model quality in a real world example about engine calibration for dynamic driving and two synthetic examples about drift and seasonal changes, see Section 5.

- (T-)IMSPE involves the computation of an integral. We consider computability of (T-)IMSPE in closed form via antiderivatives and provide a cookbook of usage for various Gaussian processes kernels and domains to be used in (T-)IMSPE, see Section 4.

- We distinctly delineate the theoretical scope of closed-form integrals and show the boundaries of practical supremacy of T-IMSPE in various delineating scenarios in Appendix D.

## 2 Background

Active learning applications typically use Gaussian process (GP) models due to their Bayesian capabilities in dealing with few data points and their reliable uncertainty estimates through variances.

### 2.1 Gaussian processes (GPs)

A *Gaussian process (GP)* $g = \mathcal{GP}(m, k)$ defines a probability distribution on the evaluations of functions $D \to \mathbb{R}$ where the domain $D \subseteq \mathbb{R}^d \cong \mathbb{R}^{1 \times d}$ such that function values $g(x_1), \ldots, g(x_n)$ at points $x_1, \ldots, x_n \in D$ are jointly (multivariate) Gaussian. Such a GP $g = \mathcal{GP}(m, k)$ is specified by a mean function $m : D \to \mathbb{R} : x \mapsto E(g(x))$ and a positive semidefinite covariance function

$$k : D^2 \to \mathbb{R} : (x, w) \mapsto E\left((g(x) - m(x))(g(w) - m(w))\right).$$

Our experiments use the kind-of-default squared exponential covariance function $k_{SE} : (x, x') \mapsto \exp\left(-\frac{||x - x'||_2^2}{2}\right)$. Any finite list of evaluations of $g$ at $x = [x_1, \ldots, x_n]$ follows the multivariate Gaussian distribution $g(x) \sim \mathcal{N}(m(x), k(x, x))$ for $g(x), m(x) \in \mathbb{R}^n$ and $k(x, x) \in \mathbb{R}^{n \times n}$ with $g(x)_i = g(x_i)$, $m(x)_i = m(x_i)$, and $k(x, x)_{i,j} = k(x_i, x_j)$. Likelihoods and posteriors can now be computed via linear algebra from this multivariate Gaussian (Rasmussen & Williams, 2006). We denote the posterior covariance function of a GP conditioned on a dataset $(x, y) \in (D \times \mathbb{R})^n$ by $k(-, -|x)$ and we define $k(x) = k(x, x)$ and $k(x, w)_{i,j} = k(x_i, w_j)$ for $w = [w_1, \ldots, w_m]$. The posterior covariance is independent of the output data $y$, which is used in active learning to plan the input $x$.

GPs are used to model dynamical systems in various ways. This paper uses NX structures in GP models (Chen & Huang, 2015; Crespi, 2019). Additionally, GPs have been used to model state space systems (Frigola et al., 2014; Eleftheriadis et al., 2017), linear differential equations (Besginow & Lange-Hegermann, 2022; Harkonen et al., 2023), or periodic dynamics (Klenske et al., 2015).

### 2.2 Time varying systems

We are motivated by learning certain time varying systems. For example, we consider systems with a behavior $y$ depending on inputs $x \in D$ in addition to drift or other time dependent changes e.g. in daily or yearly cycles. Such behavior $f$ is often modelled by including the time input $t$ amongst the model inputs.

$$y = f(x, t). \tag{1}$$

In these systems, the input $x$ can be chosen arbitrarily in an active learning approach, whereas the time $t$ is predetermined.

Another imporant kind of time varying systems is behavior $y$ where the current behavior depends on current and previous inputs. A typical example is the modeling in the calibration of engines, whose temperature not only depends on current engine speed and engine load, but on previous values of speed and load as well, e.g. in case of braking or speeding up. Other applications lie in calibration of engineering machinery like power turbines or yield prediction depending on fertilization in agriculture. Such behavior is often modelled by Nonlinear eXogenous (NX) structures, where in total $\ell$ inputs are are used, often the current input $x_t$ and the $\ell - 1$ previous inputs $x_{t-1}, x_{t-2}, \ldots, x_{t-\ell+1}$:

$$y = f(x_t, x_{t-1}, x_{t-2}, \ldots, x_{t-\ell+1}). \tag{2}$$

In such systems, the input $x_t$ can be chosen arbitrarily, whereas the inputs $x_{t-1}, x_{t-2}, \ldots, x_{t-\ell+1}$ are predetermined from previous choices.

In contrast, in dynamical systems, the combinations of state action cannot be chosen freely, as the system has to evolve to reach a certain state $x_t$. In our time varying systems, we have complete freedom to choose parts of the inputs, i.e. $x$ in (1) respectively $x_t$ in (2), and the influence of the time comes from uncontrollable conditions respectively a memory of the system that influences the system output.

Optimizing such systems (1) or (2) with Bayesian optimization is common, see e.g. (Brunzema et al., 2022). The goal of this paper is the construction of a novel acquisition function for safe active learning in such dynamic systems, including the computation of closed form integrals and a cookbook for which GP covariance functions this is possible.

## 2.3 Safe active learning

Safe Active Learning (SAL) (Schreiter et al., 2015; Zimmer et al., 2018) is a sequential experimental design which strives for safety under previously unknown constraints. SAL iteratively determines a new point $x_*$ for labelling that maximizes information $I(x_*)$, where $I$ is the so-called acquisition function, and is safe with high probability. Entropy is commonly used as a measure for information and is in case of Gaussian distribution proportional to the predictive variance of the regression GP. For a comparison and elaboration of entropy, e.g. its other names in the literature, see Subsection 3.1.

We consider the case of a safety critical quantity, e.g. a temperature should stay below a critical temperature. Then, one can calculate a safety indicator $z$ that indicates safety for non-negative values $z \geq 0$. We assume that the safety critical quantity cannot be computed, but (potentially noisy) measurements on this safety critical quantity can be obtained. Then, a safety GP $g_{\mathrm{safe}} = \mathcal{GP}(m_{\mathrm{safe}}, k_{\mathrm{safe}})$ modeling the values of $z \sim g_{\mathrm{safe}}(x_*)$ can be trained. Often but not necessary, the safety GP is the (or amongst the) model(s) that we strive to learn. This yield the probability of safely obtaining a new point $x_*$ when already some data $(x, y)$ has been obtained as

$$\xi(x_*) = \int_{z \geq 0} \mathcal{N}(z; m_{\mathrm{safe}}(x_*|x, y), k_{\mathrm{safe}}(x_*, x_*|x)) \, \mathrm{d}z.$$

Now, a new point $x_*$ can be determined by maximizing (cf. Appendix F) information constrained on safety:

$$x_* = \mathrm{argmax}_{x_* \in D, \xi(x_*) > \alpha} I(x_*) \tag{3}$$

where $0 \leq \alpha < 1$ indicates the desired safety level. The goal of this paper is the introduction of a new acquisition function for measuring information $I$ for time varying systems.

*Remark* 2.1. Our experiments use $\alpha = 0.977$ such that two standard deviations from the mean of a Gaussian is predicted as being safe. This safety criterion corresponds to the probability of a random and statistically independently drawn point being safe. In active learning, points are not drawn independently, since the data points are not drawn independently but chosen by an acquisition function. Due to this lack of statistical independence, points might have a higher probability of being unsafe. To describe our experiments, we chose the wording "satisfying" for safety percentages around 97.7 % or higher and "acceptable" for slightly lower percentages that are still above 95 %. All experiments, independent of the acquisition functions, show similar and at least acceptable safety.

After a new label $y_*$ has been obtained, both the regression and the safety GP can be retrained and the next constrained optimization problem (3) solved. More formally, SAL can be summarized in the following Algorithm 1.

We use GP hyperparameter training, which is fast and well-established. If in certain real-time applications further speed up is required, techniques of amortized inference are at hand (Bitzer et al., 2023; Liu et al., 2020). Changing hyperparameters has a strong effect on the overall performance of all algorithms, including on safety (Fiedler et al., 2024) due to model misspecification, in particular in case of few data points. Still, changing hyperparameters is state of the art in active learning and Bayesian optimization (Garnett, 2023) and, in our experience, superior to fixing hyperparameters.

**Input:** Budget $n$, initial data $(x, y) = (x_i, y_i)_{i=1}^{n_{\text{initial}}}$
**Output:** Trained Gaussian Processes
Train initial GPs;
**for** $i = 1$ **to** $n$ **do**
  Determine $x_*$ by solving (3) and get label $y_*$ for $x_*$;
  Update $(x, y) \leftarrow ((x, y), (x_*, y_*))$ and retrain GPs;
**end**

**Algorithm 1:** Safe Active Learning (SAL) with Gaussian Processes

### 2.4 Integrated Mean Squared Prediction Error (IMSPE)

The *Integrated mean squared prediction error (IMSPE)* acquisition function strives for a *model posterior with minimal average variance.* Start with a (prior) Gaussian process $g \sim \mathcal{GP}(m, k)$ and have already measured at $n$ positions $x \in \mathbb{R}^{n \times d}$. IMSPE selects $x_* \in \mathbb{R}^{1 \times d}$ with minimal

$$\int_{\mathbb{R}^d} k(x_r | x_*, x) \, d\mu(x_r), \tag{4}$$

where $k(x_r | x_*, x)$ is the posterior variance of the Gaussian processes $g$ when conditioned on observations at $x_*$ and $x$, and $\mu$ is a suitable finite measure used to average over reference points $x_r$.

Consider IMSPE for the special case of the squared exponential covariance function with lengthscale of one and signal variance of one and a Gaussian $\mu$ with mean zero and variance 1. Assume furthermore one existing measurement point at $x = 1$. Now let us consider which point $x_* \in \mathbb{R}$ to choose next, according to IMSPE. In the appendix in Example A.1, we explicitly compute the ISMPE acquisition function $\int_{\mathbb{R}} k(x_r | x_*, 1) \, d\mu(x_r)$, plotted in Figure 2. Values for $x_*$ are most suitable to reduce the variance of the posterior GP in the area around zero with variance 1 (as specified by the measure $\mu$) if they are slightly negative. The positive area is less suitable, since there already exists a measurement point $x$ at $x = 1$.

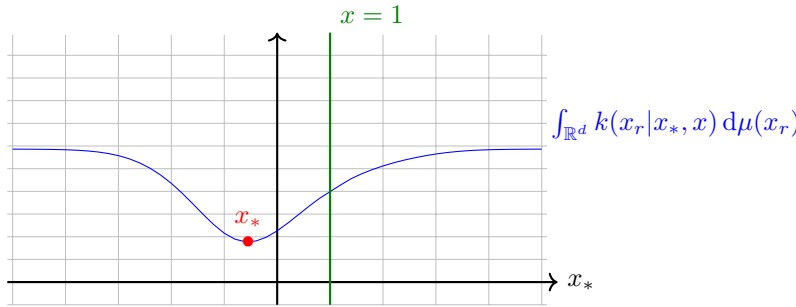

Figure 2: The IMSPE criterion in the motivating example. Without the (green) data at $x = 1$, the IMSPE criterion would be symmetric around zero. With the (green) data at $x = 1$, the optimal choice according to IMSPE for the (red) next measurement point $x_*$ is the red position at $x_* = -0.5479204538$.

We give a short discussion on IMSPE and entropy as aquisition functions for safe active learning without time awareness. We introduce related work on IMSPE in Section 3 and for further theoretical comparisons of IMSPE against other acquisition functions, we refer to Appendix D. In the realm of statistics, IMSPE stands out as the state of the art, where the superior theoretical properties of IMSPE are highly regarded. In contrast, in the field of machine learning, entropy takes precedence as state of the art. Our preliminary experiments have indicated that both communities make valid choices, and the divergence in state-of-the-art criteria arises from varied applications of GPs. In statistical communities, there is a tendency to maintain fixed GP hyperparameters in active learning methods, while in machine learning, hyperparameters are regularly optimized. Our experiments on time-varying systems, where we do optimize hyperparameters, contradict these preliminary experiments, as IMSPE is superior over entropy. In contrast to this, the ablation experiment in Appendix E.1 shows that without seasonal change ($a = 0$), entropy outperforms T-IMSPE,

which is identical to IMSPE for such time-constant systems. We note that these findings are based on initial experiments, lack scientific rigor, need a thorough follow-up, and are out of scope of this paper.

## 3 Related work

### 3.1 Acquisition functions for active learning

In machine learning, the standard acquisition function is entropy. This approach is often called ALM (Active Learning Mackay) (MacKay, 1992). The acquisition function BALD (Houlsby et al., 2011) was originally presented for classification tasks. For regression tasks, it is equivalent to the entropy criterion (Riis et al., 2022a, A.3) (Riis et al., 2022b, G.2).

The acquisition function mutual information has gained prominence in safe active learning scenarios. Originally applied in the context of sensor placement for fixed sensors (Krause et al., 2008), mutual information-based acquisition functions have found applications in active learning scenarios with discrete domain $D$ (Vasisht et al., 2014; Kirsch et al., 2019; Li et al., 2022b). For a detailed comparison of IMSPE with averaging mutual information (Krause et al., 2008) and $\beta$-diversity (Leinster, 2021), refer to Appendix D.2.

While classical criteria from the Design of Experiments (DoE) theory (Smith, 1918; Pukelsheim, 2006), such as $D$-optimal designs, could be potential acquisition functions, they have not gained traction in machine learning due to their emphasis on learning parameters rather than making predictions. IMSPE is closely connected to the classical V-optimality criterion in DoE theory, where measurement points are chosen to minimize the model's variance at a finite number of points (Fedorov & Hackl, 1997, (2.1.18)). IMSPE can be considered a variant of A-optional designs, where it minimizes the average variance of the response, while A-optimality minimizes the average variance of the model parameters. Other geometric criteria have also been proposed in practice (Thewes et al., 2016). None of these DoE criteria allow to include safety constraints.

IMSPE has a long history. The seminal work by (Sacks et al., 1989b), see also (Sacks et al., 1989a) and the authors' book (Santner et al., 2003), introduced the acquisition function IMSE (Integrated Mean Squared Error), estimating mean squared error via posterior variance for polynomial covariance functions. (Ying, 1991) renamed IMSE into IMSPE and conducts extensive theoretical investigations into its asymptotic properties. The second time the name IMSPE appears in the literature is in (Fang, 2000). (Ankenman et al., 2010) proposes the use of IMSE, for which they use a closed form formula, for active learning. (Leatherman et al., 2014) gives a thorough overview about techniques for static designs with IMSPE. (Leatherman et al., 2018) computes the IMSPE for certain finite dimensional GPs. For modern references to IMSPE we refer to (Gramacy, 2020; Binois et al., 2019), with extensive implementation hetGP (Binois & Gramacy, 2021) in the system R.

IMSPE has been reintroduced under different names. (Cohn et al., 1996) introduced active learning for mixtures of Gaussian with the goal of minimizing IV (Integrated Variance), for which they found a simple closed form; this is another special case of IMSPE later named ALC[1] (Active Learning Cohn). (Seo et al., 2000) introduced active learning to GPs, where they used VI from ALC while approximating the integral using sampled points. (Burnaev & Panov, 2015) independently introduced IMSPE as IntegratedMSEGain, providing an unproven closed form. (Gorodetsky & Marzouk, 2016) independently introduce IMSPE under the name IVAR (Integrated VARiance), where they compute the integral numerically. Recent literature has extensively utilized IMSPE or its variants. (Vernon et al., 2019) focuses on safety boundaries, (Meka et al., 2020) applies ALC in closed-set active learning, (Lee et al., 2022) investigates IMSE, ALC, and ALM in safe active learning, and (Lee et al., 2023) employs a discrete approximation to ALC/IMSE for local GPs.

None of these approaches use IMSPE or its variants under different names to specifically collect information for future time steps.

---

[1]Some authors distinguish between IMSPE and ALC (Sauer et al., 2023), noting that IMSPE computes the integral in equation (4) in closed form, while ALC approximates it numerically. However, this distinction is not widely recognized in the literature. For an example, see (Gramacy, 2020, §6) by the second author of (Sauer et al., 2023).

### 3.2 Active learning in dynamic systems

Active learning has also been explored in dynamic systems. (Schneider, 1995; Umlauft et al., 2020) consider dropping old data points when their information content becomes irrelevant. (Lughofer, 2017) provides an overview of active learning in data streams, while (Jain et al., 2018) applies active learning to GP models for non-linear control during closed-loop control. Safety considerations in dynamic systems are addressed by (Zimmer et al., 2018), and (Buisson-Fenet et al., 2020) proposes an iterative active learning scheme in dynamical systems based on mutual information. (Yu et al., 2021) conducts active learning specifically for GP state space models, and (Heim et al., 2020) learns safety constraints in dynamical systems. Similar strategies can be found in reinforcement learning for non-stationary environments (Padakandla et al., 2020; Padakandla, 2021; Ritto et al., 2022; Cowen-Rivers et al., 2022), control (Fisac et al., 2018; Agarwal et al., 2019; Capone et al., 2020), concept drift (Han et al., 2022), and theoretical work on regret in non-stationary systems (Zhang, 2021; Zhao et al., 2022). The papers (Fiducioso et al., 2019; Krause & Ong, 2011) consider regret in the current time/context and show that their choice of acquisition function is still sublinear in changing contexts.

Notably, none of these approaches considers acquisition functions that account for information or prediction accuracy at future time steps.

### 3.3 Safe learning

Safe exploration has been employed in robotics (Sui et al., 2018; Berkenkamp et al., 2016; Baumann et al., 2021), energy management (Galichet et al., 2013), terrain exploration, (Moldovan & Abbeel, 2012; Turchetta et al., 2019) and engine modeling (Schreiter et al., 2015; Zimmer et al., 2018; Schillinger et al., 2017; Li et al., 2022a). While most of this Safe Learning work is on Bayesian optimization, some is on active learning but does not consider future information or future prediction accuracy. (Li et al., 2024) considers Bayesian optimization (but not active learning) in a dynamic setting, with a focus on safety guarantees.

Active learning aims at learning a model over the whole input space to perform some post-hoc or worst-case analysis. In contrast, Bayesian optimization is only interested in an optimal point of operation. Once this point or the area is found, the rest of the input space is not of any interest anymore. In this regard, active learning and Bayesian optimization are conceptually different. Specifically, IMSPE reduces the variance globally and seems unreasonable to be integrated meaningfully into a Bayesian optimization framework.

## 4 T-IMSPE - Time-aware Integrated Mean Squared Prediction Error

Our criterion T-IMSPE uses the concepts and formulas from IMSPE in time-dependent models, to ensure variance reduction not only for the current time step, but also for future time steps. We first consider GP models with time as an additional input for both continuous and discrete time domains, and afterwards we consider NX-GP models for dynamic systems in discrete time domain. In both cases, T-IMSPE work with a wide range of GP covariance structures.

### 4.1 T-IMSPE for GPs with time amongst its inputs

Here, we consider time varying systems as in (1), where the time is one of the model inputs. Consider a GP $g = \mathcal{GP}(m, k)$ defined on the domain $\mathbb{R} \times D$, where $\mathbb{R}$ represents the time domain and $D \subseteq \mathbb{R}^d$ represents the remaining inputs, e.g. "spatial" inputs. For example, when not encoding any specific time dependent behavior in the GP prior, $k$ might be a squared exponential covariance function on $\mathbb{R}^{d+1}$. Let $\mu$ be a finite measure on $\mathbb{R} \times D$. We define T-IMPSE for choosing a new data point $x_* \in D$ at time $t_* \in \mathbb{R}$ after already obtained data $(\tau, x) \in (\mathbb{R} \times D)^n$ for such GPs as

$$\text{T-IMSPE}(x_*) = \int k((t_r, x_r)|(t_*, x_*), (\tau, x)) \, \mathrm{d}\mu(t_r, x_r). \tag{5}$$

By Theorem 4.4 below, this integral is computable in closed form for most common covariance functions. The time-aware aspect can now be included in the measure $\mu$. Assume we have a suitable measure $\mu_D$ on the

domain $D$. Then, we can encode our desire to accumulate information in the relevant time interval $[t_*, t_* + \Delta t]$ by a uniform distribution $\mu_t = \mathbf{1}_{[t_*, t_* + \Delta t]}$. Then, defining $\mu$ as the product measure $\mu := \mu_t \otimes \mu_D$ strives to choose a new data point as follow: the posterior variance over the domain $D$ is reduced being weighed by $\mu_D$, while this reduction is not only achieved at $t_*$, but aware of future times in all of $[t_*, t_* + \Delta t]$. The relevant time domain might also be considered discrete via $\mu_t = \mathbf{1}_{\{t_*, t_*+1, \ldots, t_*+\Delta t\}}$, then $\mu := \mu_t \otimes \mu_D = \sum_{t=t_*}^{t_*+\Delta t} \delta_t \otimes \mu_D$ for the Kronecker delta $\delta_t$.

## 4.2   T-IMSPE for GPs with NX-structure

Here, we consider time varying systems as in (2), where the time dependency is modeled via an NX structure. Consider $y_t = f(\hat{x}_t, \hat{x}_{t-1}, \ldots, \hat{x}_{t-\ell+1})$ for some positive $\ell \in \mathbb{N}$ by placing a GP prior $g = \mathcal{GP}(m, k)$ on the function $f : D^\ell \to \mathbb{R}$. Such models work in discrete time domain. Let $\mu$ be a finite measure defined on $D^\ell$. Assume we already obtained $n$ data points at $x \in (D^\ell)^n$, where the condition $x_{i,t} = x_{i+1,t+1} \in D$ for the NX structure for $i \in \{1, 2, \ldots, \ell-1\}$ and $t \in \{1, 2, \ldots, n-1\}$. We define T-IMPSE for choosing a new data point $x_* \in D$ for such NX GPs as

$$\text{T-IMSPE}(x_*) = \int k(x_r | \overline{x_*}, x) \, \mathrm{d}\mu(x_r), \tag{6}$$

where $\overline{x_*} = (x_*, x_{1,n}, \ldots, x_{\ell-1,n}) \in D^\ell$ is the new measurement position with previous measurement positions to predict the model dynamics. Since $x_{1,n}, \ldots, x_{\ell-1,n}$ are the measurement positions from previous steps, they are well known values, to which we add the vector $x_*$ of new inputs. We now use the closed formula for T-IMSPE in $\int k(x_r \mid x_*, x) d\mu(x_r)$ from Theorem 4.4, where instead of $x_*$ we substitute $\overline{x_*}$. This formula allows optimization for $x_*$.

In our experiments we chose the $\ell$-times product measure $\mu := \bigotimes_{i=1}^\ell \mu_D$ for a fixed finite measure $\mu_D$ on the domain $D$. Note that if $\mu_D$ is Gaussian or a uniform measure on a multi-dimensional interval, then so is $\mu$. This acquisition function is time-aware, as points are chosen that reduce the variance over all possible dynamical behaviour.

This version of T-IMSPE is a special case of the previous one: start with the version from Subsection 4.1, ignore the time dependency in the GP and intepret the GP to have the NX-time inputs. Now, the additional average over time in T-IMPSE is trivial.

## 4.3   $P$-elementary functions for closed form (T-)IMSPE

A key step to applicability of our new T-IMSPE criterion is the evaluation of the integral in Equation (4), and hence in the Equations (5) and (6). This section shows that the integral can be computed in closed form for a variety of covariance functions and measures. This decreases computational load improves numerical stability of values and gradient. Even though this work uses the squared exponential kernel, work on efficient kernel selection (Bitzer et al., 2022) allows the modeler to quickly pick a suitable composed kernel. Our theoretical contribution actually extends beyond our T-IMSPE criterion to the general situation of IMSPE. We summarize and extend existing results (Sacks et al., 1989b; Leatherman et al., 2018) that this integral can be computed in *closed form*, putting them into a more abstract and widely applicable framework.

Recall the class of elementary functions, which was defined in the 19th century (Liouville, 1833a;c;b; Bronstein, 2005) to e.g. describe the well known result that while $\mathrm{erf}(x) := \int \exp(-x^2) \, \mathrm{d}x$ is analytical and even entire, it is not expressible in closed form using classical functions. The class of elementary functions is not suitable to describe functions being computable in PyTorch, e.g. it is too big in the sense that it is closed under inverse functions and it is too small as functions like $\mathrm{erf}(x)$ are nowadays easily computable. Furthermore, the class of elementary functions only considers univariate scalar function $f : U \to \mathbb{R}$ for $U \subseteq \mathbb{R}$, whereas multivariate multi-ouput functions are common in machine learning. Now, we adapt this definition of elementary functions to formally define what it means to compute functions in closed form in a system like PyTorch.

Fix a programming framework $P$. We think of $P$ as PyTorch, but one might as well think of $P$ as JAX, TF or any other current or future framework. Then, we view the following operation as computable in closed

form in $P$: (a) they have access to functions in $P$, (b) one can compose functions, and (c) one can perform arithmetic operations. More formally, we define the following.

**Definition 4.1.** We define the set of $P$**-elementary functions** as the intersection[2] of all sets of partial[3] functions $U \to \mathbb{R}^{d''}$ for $U \subseteq \mathbb{R}^{d'}$, $d', d'' \in \mathbb{Z}_{>0}$, that

(a) include all constants (considered as constant functions) and functions available in $P$,

(b) are closed under function composition $\circ$, and

(c) are closed under (multivariate and multi-output) rational maps; in particular additions, subtractions, multiplications, divisions, and linear maps.

**Example 4.2.** The following functions are PyTorch-elementary:

i. $(x_1, x_2) \mapsto 2x_1 - 42x_2$ by Definition 4.1.(c).

ii. $x_1 \mapsto \mathrm{erf}(x_1)$ by Definition 4.1.(a).

iii. $(x_1, x_2) \mapsto \mathrm{erf}(x_1)$ by using all of Definition 4.1, as $(x_1 \mapsto \mathrm{erf}(x_1)) \circ ((x_1, x_2) \mapsto x_1)$.

iv. $(x_1, x_2, x_3) \mapsto \mathrm{erf}(\mathrm{softmax}(x_1 + \pi \cdot x_2, \Gamma(\frac{x_3}{x_2})^2))$ by using all of Definition 4.1.

We now show that IMSPE and T-IMSPE are PyTorch-elementary for a wide range of covariance functions, making safe active learning with them is computationally efficient and numerically stable.

## 4.4 Computability in closed form

IMSPE and T-IMSPE can be computed in closed form with a mild assumption on the GP covariance.

**Definition 4.3.** Let $\mu$ be a finite measure on $\mathbb{R}^d$. We say a covariance function $k$ is $P$**-elementary covariance marginalizable** w.r.t. $\mu$ if the integral $\int k(x_1, x_r) k(x_r, x_2) \, \mathrm{d}\mu(x_r)$ exists as a $P$-elementary function in $x_1$ and $x_2$.

This is a non-trivial definition. In the appendices, we show or recall from the existing literature that constant (Example A.3), ARD squared exponential (Examples A.4), polynomial (Example A.5), half-integer Matérn (Example A.5), Wiener process (Example A.5), and random Fourier feature (Example A.11) covariance functions are Pytorch-elementary covariance, while rational quadratic and periodic covariance functions are probably not (Example A.6), all when considering $\mu$ as a non-degenerate continuous uniform distribution on a multidimensional interval or non-degenerate Gaussian distribution.

The class of $P$-elementary covariance marginalizable covariance functions is closed under various operations: scaling (Proposition A.8), multiplication on independent inputs (Proposition A.9), and also under sums when the covariance functions satisfy another condition (Proposition A.10). Furthermore, if a covariance function is $P$-elementary covariance marginalizable w.r.t. two measures, then so it is w.r.t. the sum of these two measures (Proposition A.12). This *cookbook* allows to construct new GPs from previous ones, such that T-IMSPE stays applicable. For more details on this cookbook see Appendix A.3. From this discussion we conclude that the following theorem is widely applicable.

**Theorem 4.4.** *Assume the prior covariance function $k$ of a prior GP $g$ to be $P$-elementary covariance marginalizable w.r.t. a measure $\mu$. Then, IMSPE and T-IMSPE are $P$-elementary, i.e. they can be computed in closed form in the programming framework $P$.*

These results have the major advantage that it is sufficient for the prior covariance function to be $P$-elementary covariance marginalizable, and we do not need properties for the posterior covariance functions. For a proof

---

[2]Intersecting is a way of constructing mathematical objects when direct constructions are inconvenient. E.g., the span of a set of vectors can be defined as intersection of all linear subspaces containing this set of vectors.

[3]Partial functions are functions that are not defined in all of their domain. This is necessary, e.g. as we divide through functions having zeros or use logarithms.

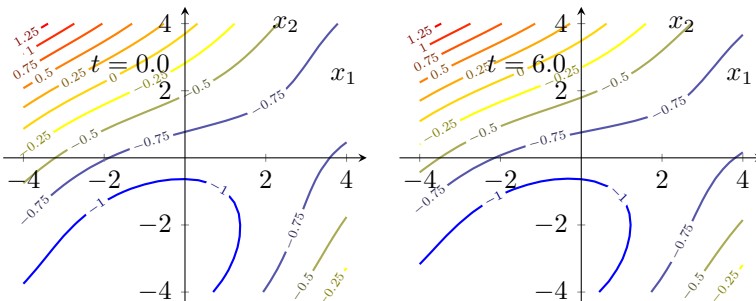

Figure 3: Plots for the seasonal change experiment (Subsection 5.1) for $t = 0$ (left) and $t = 6$ (right).

of Theorem 4.4 we refer to Appendix B, which also provides a very concrete formula for the computation of these criteria. We do not claim novelty to these proofs, which are known for the special case of polynomial covariance functions since (Sacks et al., 1989b) and additional special cases are spread over the subsequent literature, see Subsection 3.1.

We follow the state of the art approach in active learning, where GPs are trained after each addition of a new data point, including hyperparameter training. After this training, IMSPE and T-IMSPE have the same computational complexity as computing the entropy and can be evaluted on $O(n^2)$ for $n$ training data points.

**Corollary 4.5.** *IMSPE and T-IMSPE as computed in the proof of Theorem 4.4 need a one-time Cholesky decomposition (which is computed anyway during the GP training) of the data covariance matrix $k(x, x)$ in $O(n^3)$, independent of the new data point $x_*$, where $n$ is the number of data points. Afterwards, these criteria can be evaluated in $O(n^2)$ as a function in $x_*$.*

The proof of this corollary in Appendix B is very explicit and yields a direct formula for computations. For us, the most important special case of the above general theory is the following.

**Corollary 4.6.** *IMSPE and T-IMSPE can be computed in closed form in PyTorch for a prior GP with squared exponential covariance, provided the measure $\mu$ is Gaussian or continuous uniform.*

## 5 Experiments

We conduct three experiments for safe active learning (Algorithm 1) in time varying systems. Thereby, we demonstrate the superiority of the modeling quality achieved by T-IMSPE over both entropy, which is currently the state of the art (see Appendix D) in active learning, and IMSPE, while keeping the same safety standard. For further results of our experiments resp. further details on the setup of our experiments see Appendix E resp. Appendix F and the attached code. We use the paired Wilcox signed rank test to show statistical significance.

### 5.1 Experiment: seasonal change

We consider learning a system with strong periodic seasonal changes. The system is given as by rotating the two-dimensional domain in the function from Equation (7) in Appendix F.1 and plotted in Figure 3 with additional plots in the Appendix in Figure 11. It is particular challenging due to the quick period in time of $4\pi \approx 12.6$ and high variation of around 50 % in its range. These changes also strongly affect the position of the safe area.

We start with 8 initial measurements at times $0, \ldots, 7$ positioned at the inital points of a Sobol sequence in the safe area. Afterwards, 100 further measurements at times $8, \ldots, 107$ are conducted according to the respective safe active learning criteria T-IMSPE (from Equation 5), entropy, and IMSPE.

For the upcoming experiment and the subsequent experiment, we utilize a grid as our test dataset, restricting to grid points where the behavior is currently safe. The rationale behind restricting our test data to this safe area is rooted assessing the model's quality there. The test data varies across different time steps, reflecting

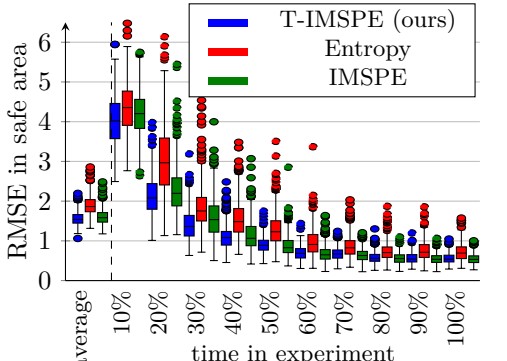 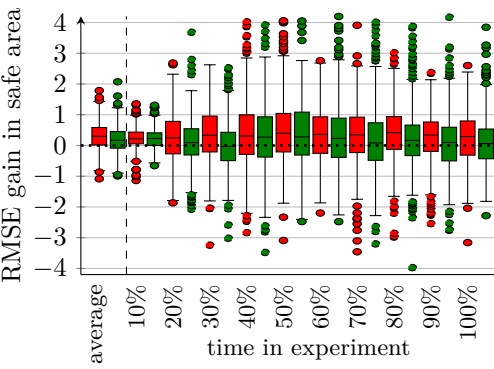

Figure 4: Box plots of RMSE value in the safe area of the experiments from active learning with seasonal changes (left, Subsection 5.1) and under drift (right, Subsection 5.2). The x-axis shows these value on averaged over all time steps and at certain ascending time steps from 10% to 100%. The left diagram compares the results of 450 runs between T-IMSPE (blue), entropy (red), and IMSPE (green), where T-IMSPE is highly significantly ($p < 2.2\text{e}-16$) superior to entropy in all eleven comparison, whereas T-IMSPE is superior over IMSPE on average ($p < 2.1\text{e}-3$). The median error reached by T-IMSPE at 30 % of the time is reached by entropy at 50 % of the time, which is a 40 % reduction of measuremt time and cost. The right diagram shows the gain of RMSE values in 500 runs when choosing T-IMSPE over entropy (red) and over IMSPE (green). The results are significantly ($p < 1\text{e}-10$) better for T-IMSPE over entropy in all 11 comparisons and and significantly better in 10 out of 11 comparisons for T-IMSPE over IMSPE.

the changing state of the system. Consequently, our analysis is confined to the current time, deliberately excluding future time steps. This intentional exclusion prevents any unwarranted advantage of T-IMSPE over entropy in the evaluation criteria, as T-IMSPE optimizes over future time steps. This precaution ensures a fair and unbiased evaluation of model performance within the immediate temporal context. Comparisons are done between runs of equal random seeds, in particular the same noise is added to the initial measurement points.

Due to changing domains, we strive for a good RMSE not only at the final time step, but during all of the measurement time. Figure 4 summarizes the results in terms RMSE in 50 runs with different seeds by the above protocol after 10, 20, ..., 100 time steps, and the average over all time steps. In this experiment, T-IMSPE is vastly superior in reducing RMSE in comparison to the state of the art entropy. The superiority of T-IMSPE is highly significant ($p < 2.2\text{e}-16$) for all time steps and the average model quality. This number $2.2\text{e}-16$ is the smallest representable $p$-value in R and indicates a result that is statistically highly significant, essentially suggesting that the probability of the observed result occurring under the null hypothesis is vanishingly small. Comparing T-IMSPE to IMSPE, we have T-IMSPE being superior to IMSPE on average ($p < 2.1\text{e}-3$) and over the first three time steps at 10 % to 30 % ($p < 2.2\text{e}-4$, $p < 1.1\text{e}-2$, $p < 1.5\text{e}-7$). Afterwards, when IMSPE and T-IMSPE have both mostly converged, both methods perform comparably, with advantages for IMSPE at time steps 40 % to 90 %.

With all acquisition functions, over 99.5 % of all points were safe; this is satisfying by Remark 2.1.

## 5.2 Experiment: drift

We consider learning a system with strong temporal drift. The system is given as by the formula in Equation (8) in Appendix F.1 and plotted in Figure 5 with additional plots in the Appendix in Figure 12 and Figure 13. This system is particularly challenging due to the strong drift effect: the safe area shrinks in size by approximately a factor of 2 and the range of the function increases by a factor of more than 150. With the exception of the changed formula, our setup is the same as for the experiment with seasonal drift in the previous Subsection 5.1.

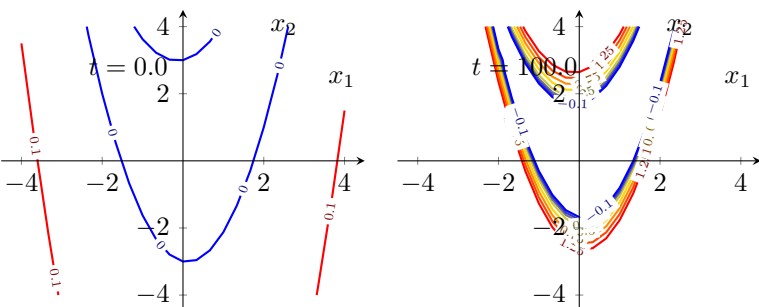

Figure 5: Plots of the drift experiment (Subsection 5.2) for $t = 0$ (left) and $t = 100$ (right).

Due to changing domains, we strive for a good RMSE not only at the final time step, but during all of the measurement time. Figure 4 summarizes the results in terms RMSE in 500 runs with different seeds by the above protocol after 10, 20, . . . , 100 time steps, and the average over all time steps. We see that at almost all time steps, and in particular on average (left), the RMSE of T-IMSPE is smaller than that of the entropy. In a

comparisons. The superiority of T-IMSPE over IMSPE is highly significant ($p < 2e{-}16$) on average, highly significant ($p < 1e{-}5$) for 5 time steps (10 %, 40 %, 50 %, 60 %, 80 %), significant ($p < 5e{-}2$) for 4 time steps (20 %, 70 %, 90 %, 100 %), and inconclusive at one time step (30 %).

We have 96.2 %–96.5 % safe points for all 3 acquisition functions; this is acceptable by Remark 2.1.

### 5.3 Dynamic real world system: rail pressure

This experiment considers the rail pressure example from (Tietze et al., 2014). The system consists of an actuation $v_k$ and an engine (rotational) speed $n_k$, which yield the rail pressure $\psi_k$. This system is particularly challenging, as measurement positions at one time step strongly affect the measurements at subsequent time steps. Furthermore, such measurements are quite expensive in real world scenarios. There exists a formula, given in the code, such that $\psi_k = \psi(n_k, n_{k-1}, n_{k-2}, n_{k-3}, v_k, v_{k-1}, v_{k-3})$ with $v_{k-2}$ missing, and we will try to find a GP model $g$ such that $\psi_k \approx g(n_k, n_{k-1}, n_{k-2}, n_{k-3}, v_k, v_{k-1}, v_{k-2}, v_{k-3})$. Here, the final RMSE values in the safe domain are most important in practice. To obtain test data in the safe area, we constructed a random safe trajectory of length 2024, where the next point is always a random point that turned out to be safe.

Previous papers considered this dynamic example using piecewise linear trajectories as in (Zimmer et al., 2018), where given a history of measurement position $(n_k, v_k), (n_{k-1}, v_{k-1}), \ldots,$ a new point $(n_{k+5}, v_{k+5})$ was chosen and the measurement position between were linearly interpolated as $(n_{k+i}, v_{k+i}) = \frac{5-i}{5}(n_k, v_k) + \frac{i}{5}(n_{k+5}, v_{k+5})$. We consider fully dynamic learning, where given the above history, $(n_{k+1}, v_{k+1})$ is chosen. Hence, we are in the setting of NX models as in Subsection 4.2, where IMSPE is not feasible as a comparison.

The results of the RMSE of comparing 100 runs of entropy to 100 runs of T-IMSPE (from Equation 6) over 1000 iterations are shown in Figure 6. The superiority in RMSE values shows the superiority of the T-IMSPE over entropy for dynamical systems. The Wilcox signed rank test conducted 10 times (every 100 steps) shows that at all time steps TIMSPE is significantly better than entropy with $p < 1.1e{-}6$ after 100 time steps and $p < 3.2e{-}15$ (sic!) at all remaining tested time steps.

Entropy had 97.5 % safe points and T-IMSPE had 99.2 % safe points in these experiments, see also Table 2 in Appendix E, which is satisfying for both acquisition functions by Remark 2.1.

## 6 Conclusion

This paper introduced an algorithm for safe active learning that is specifically suited for time varying systems. Our T-IMSPE acquisition function is able to capture the time dependence and does not only collect information for the current time step, but also for future time steps.

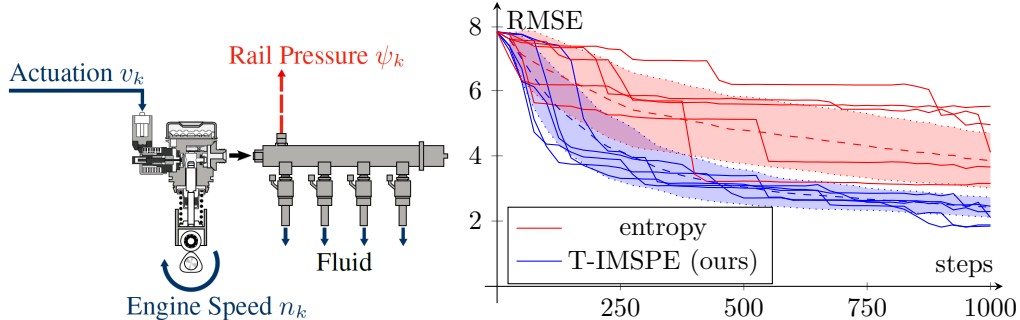

Figure 6: Left: Common rail pressure system (system for high-pressure fuel injection) with controllable inputs $v_k, n_k$ and measured output $\psi_k$, taken from Zimmer et al. (2018); Tietze et al. (2014) Right: This diagram show the decline in RMSE in the rail pressure model from Subsection 5.3. Entropy (red) shows a slow decline over the 1000 steps, whereas our approach T-IMSPE declines more consistensly, faster, and ends in much smaller RMSE values. The dashed line is the mean of 100 runs, the area shows the $2\sigma$ area and the solid lines show 5 examplatory runs. The mean error reached by T-IMSE in the rail pressure example after 250 steps is the same as the one reached by entropy after 1000 steps. This is an 75 % reduction of measurement time and cost

Our T-IMSPE acquisition function can be evaluated in closed-form at comparable complexity as the widely used entropy acquisition function. This holds for commonly used kernels and we theoretically show the boundary of computability for the IMSPE and T-IMSPE criteria and construct a cookbook of GP constructions for IMSPE and T-IMSPE. The superiority of TIMSPE is highly significant.

Additionally, the theoretical contributions of this paper allow further applications which need more control over information collection; for example one can specify where and when information is important and one can see for which covariance functions such information gathering is possible.

The experimental results show a clear and significant advantage for the model quality obtained by T-IMSPE over the state of the art entropy and over IMSPE in various settings of time changing systems, while keeping the safety. In Appendix E.1 we discuss what happens if systems are only slightly time changing.

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

# A  On $P$-elementary covariance marginalizable covariance functions

This section provides more information about $P$-elementary covariance marginalizable covariance functions. In particular, we provide a cookbook of which GP covariance constructions result in $P$-elementary covariance marginalizable covariance functions.

## A.1  Motivating example

Recall Definition 4.3: a covariance function $k$ is $P$-elementary covariance marginalizable w.r.t. a finite measure $\mu$ on $\mathbb{R}^d$ if the integral $\int k(x_1, x_r)k(x_r, x_2)\, \mathrm{d}\mu(x_r)$ exists as a $P$-elementary function in $x_1$ and $x_2$. This condition was used in Theorem 4.4, and hence allowed for closed form computations of IMSPE and T-IMSPE.

Let us see this definition in action for the special case of the squared exponential covariance function with lengthscale of one and signal variance of one and a Gaussian measure, as already considered in Subsection 2.4. The rest of this appendix section vastly generalizes this example.

**Example A.1.** According to Formula (4), IMSPE selects the next measurement point $x_* \in \mathbb{R}^{1 \times d}$ such that the averaged posterior variance

$$\int_{\mathbb{R}^d} k(x_r | x_*, x)\, \mathrm{d}\mu(x_r),$$

is minimal for any chosen measure $\mu$ on $\mathbb{R}^{1 \times d}$. Consider specifically $d = 1$, the prior Gaussian process $g = \mathcal{GP}(0, k)$ with mean zero and covariance function

$$k(x_1, x_2) = \exp\left(-\frac{(x_1 - x_2)^2}{2}\right),$$

and $\mu$ the standard Gaussian distribution given by the (Lebesgue, denoted by $\lambda$) density

$$\frac{\mathrm{d}\mu}{\mathrm{d}\lambda} = \frac{1}{\sqrt{2\pi}} \exp\left(-\frac{x^2}{2}\right).$$

Assume furthermore one existing measurement point at $x = 1$.

Now let us consider which point $x_* \in \mathbb{R}$ to choose next, according to IMSPE.

$$\int_{\mathbb{R}} k(x_r | x_*, 1)\, \mathrm{d}\mu(x_r)$$

$$= \int_{\mathbb{R}} \left( k(x_r, x_r) - \begin{bmatrix} k(x_r, x_*) & k(x_r, 1) \end{bmatrix} \begin{bmatrix} k(x_*, x_*) & k(x_*, 1) \\ k(1, x_*) & k(1, 1) \end{bmatrix}^{-1} \begin{bmatrix} k(x_*, x_r) \\ k(1, x_r) \end{bmatrix} \right) \mathrm{d}\mu(x_r)$$

$$= \int_{\mathbb{R}} \left( 1 - \begin{bmatrix} k(x_r, x_*) & k(x_r, 1) \end{bmatrix} \begin{bmatrix} 1 & k(x_*, 1) \\ k(1, x_*) & 1 \end{bmatrix}^{-1} \begin{bmatrix} k(x_*, x_r) \\ k(1, x_r) \end{bmatrix} \right) \mathrm{d}\mu(x_r)$$

$$= 1 - \int_{\mathbb{R}} \left( \frac{1}{1 - k(1, x_*)^2} \begin{bmatrix} k(x_r, x_*) & k(x_r, 1) \end{bmatrix} \begin{bmatrix} 1 & -k(1, x_*) \\ -k(1, x_*) & 1 \end{bmatrix} \begin{bmatrix} k(x_*, x_r) \\ k(1, x_r) \end{bmatrix} \right) \mathrm{d}\mu(x_r)$$

$$= 1 - \frac{1}{1 - k(1, x_*)^2} \int_{\mathbb{R}} \begin{bmatrix} k(x_r, x_*) & k(x_r, 1) \end{bmatrix} \begin{bmatrix} 1 & -k(1, x_*) \\ -k(1, x_*) & 1 \end{bmatrix} \begin{bmatrix} k(x_*, x_r) \\ k(1, x_r) \end{bmatrix} \mathrm{d}\mu(x_r)$$

$$= 1 - \frac{1}{1 - k(1, x_*)^2} \int_{\mathbb{R}} \left( k(x_r, x_*)^2 + k(x_r, 1)^2 - 2k(1, x_*)k(x_r, x_*)k(x_r, 1) \right) \mathrm{d}\mu(x_r)$$

This integral is computable, if we can compute the above products (or squares) of covariance functions. Computing this is closed form is precisely the condition of $k$ being $P$-elementary covariance marginalizable w.r.t. $\mu$. Luckily, this is possible for Gaussians, see also Example A.4 for a more general statement of this fact.

$$= 1 - \frac{1}{(1 - k(1, x_*)^2)}$$
$$\left( \left( \int_{\mathbb{R}} \exp\left(-\frac{(x_r - x_*)^2}{2}\right) \right)^2 \mathrm{d}\mu(x_r) + \int_{\mathbb{R}} \exp\left(-\frac{(x_r - 1)^2}{2}\right)^2 \mathrm{d}\mu(x_r) \right.$$
$$\left. - 2 \int_{\mathbb{R}} \exp\left(-\frac{(1 - x_*)^2}{2}\right) \exp\left(-\frac{(x_r - x_*)^2}{2}\right) \exp\left(-\frac{(x_r - 1)^2}{2}\right) \mathrm{d}\mu(x_r) \right)$$

$$= 1 - \frac{1}{(1 - k(1, x_*)^2)\sqrt{2\pi}}$$
$$\left( \int_{\mathbb{R}} \exp\left(-\frac{(x_r - x_*)^2}{2}\right)^2 \exp\left(-\frac{x_r^2}{2}\right) \mathrm{d}\lambda(x_r) + \int_{\mathbb{R}} \exp\left(-\frac{(x_r - 1)^2}{2}\right)^2 \exp\left(-\frac{x_r^2}{2}\right) \mathrm{d}\lambda(x_r) \right.$$
$$\left. - 2 \exp\left(-\frac{(1 - x_*)^2}{2}\right) \int_{\mathbb{R}} \exp\left(-\frac{(x_r - x_*)^2}{2}\right) \exp\left(-\frac{(x_r - 1)^2}{2}\right) \exp\left(-\frac{x_r^2}{2}\right) \mathrm{d}\lambda(x_r) \right)$$

$$= 1 - \frac{1}{(1 - k(1, x_*)^2)\sqrt{2\pi}}$$
$$\left( \int_{\mathbb{R}} \exp\left(-(x_r - x_*)^2 - \frac{x_r^2}{2}\right) \mathrm{d}\lambda(x_r) + \int_{\mathbb{R}} \exp\left(-(x_r - 1)^2 - \frac{x_r^2}{2}\right) \mathrm{d}\lambda(x_r) \right.$$
$$\left. - 2 \exp\left(-\frac{(1 - x_*)^2}{2}\right) \int_{\mathbb{R}} \exp\left(-\frac{(x_r - x_*)^2}{2} - \frac{(x_r - 1)^2}{2} - \frac{x_r^2}{2}\right) \mathrm{d}\lambda(x_r) \right)$$

Again, we stress that these integrals are computable in closed form.

$$= 1 - \frac{1}{(1 - k(1, x_*)^2)\sqrt{2\pi}} \left( \frac{\sqrt{2\pi}}{\sqrt{3}} \exp\left(-\frac{x_*^2}{3}\right) + \frac{\sqrt{2\pi}}{\sqrt{3}} \exp\left(-\frac{1}{3}\right) - \frac{2\sqrt{2\pi}}{\sqrt{3}} \exp\left(-\frac{1}{6}(5x_*^2 - 8x_* + 5)\right) \right)$$

$$= 1 - \frac{1}{(1 - \exp(-(1 - x_*)^2))} \left( \frac{1}{\sqrt{3}} \exp\left(-\frac{x_*^2}{3}\right) + \frac{1}{\sqrt{3}} \exp\left(-\frac{1}{3}\right) - \frac{2}{\sqrt{3}} \exp\left(-\frac{1}{6}(5x_*^2 - 8x_* + 5)\right) \right)$$

The resulting acquisition function shows which values for $x_*$ are most suitable to yield information concentrated around zero with variance 1 (as specified by the measure $\mu$) for the given Gaussian process. We plot this function in Figure 2 and discuss it in Subsection 2.4.

This example has shown on the special case that we need to integrate certain products of covariance functions in closed form to compute IMSPE and T-IMSPE in closed form. This was the definition of $P$-elementary covariance marginalizable covariance functions, which will be formally used in the proof of Theorem 4.4 in Appendix B, similar to the example above.

## A.2 Examples of $P$-elementary covariance marginalizable covariance functions

We now generalize this definition of a covariance function $k$ being $P$-elementary covariance marginalizable, such that it takes two covariance functions info consideration. This generalization is necessary for taking sums of two covariance functions, see Proposition A.10. Our proofs below are given in this slightly more general framework.

**Definition A.2.** Let $\mu$ be a finite measure on $\mathbb{R}^d$. We call a pair $(k_1, k_2)$ of two covariance functions $P$-**elementary** *cross*-**covariance marginalizable** w.r.t. $\mu$ if the integral $\int k_1(x_1, x_r) k_2(x_r, x_2) \mathrm{d}\mu(x_r)$ exists as a $P$-elementary function in $x_1$ and $x_2$.

Obviously, a covariance function $k$ is $P$-elementary covariance marginalizable w.r.t. $\mu$ if $(k, k)$ is $P$-elementary cross-covariance marginalizable. Hence, we will show some of the following results for $P$-elementary cross-covariance marginalizability, which implies these results for $P$-elementary covariance marginalizability.

The definition of $P$-elementary cross-covariance marginalizability is non-trivial, as many relevant covariance functions are elementary (cross-)covariance marginalizable w.r.t. relevant measures. While we state the following examples with $P$ being PyTorch, these results should obviously hold for any reasonable programming framework.

**Example A.3.** Constant covariance functions $k_i(x_1, x_2) = c_i$ are PyTorch-elementary cross-covariance marginalizable w.r.t. any finite measure $\mu$. The integral

$$\int_D k_1(x_1, x_r)k_2(x_r, x_2)\,\mathrm{d}\mu(x_r) = \int_D c_2 c_2\,\mathrm{d}\mu(x_r)$$
$$= c_1 c_2 \cdot \mu(D) \in \mathbb{R}$$

is a real number since $\mu$ is a finite measure. Hence, this real number can be represented by PyTorch.

**Example A.4.** Consider the automatic relevance detection squared exponential (SE) covariance function

$$k_{\mathrm{SE}, \sigma, \ell}(x, y) = \sigma^2 \exp\left(-\frac{1}{2}\sum_{i=1}^{d}\frac{(x_i - y_i)^2}{\ell_i^2}\right).$$

Then, $k_{\mathrm{SE}, \sigma_1, \ell_1}$ and $k_{\mathrm{SE}, \sigma_2, \ell_2}$ are PyTorch-elementary cross-covariance marginalizable w.r.t. any continuous uniform distribution on a multidimensional interval or any Gaussian distribution. This holds, since any finite product of Gaussian functions is again Gaussian, leading to integrals involving the error function erf or more squared exponential functions. To prevent digging even deeper into gory details of indices, here we demonstrate this in the one-dimensional case for a uniform measure $\mu = c \cdot \mathbf{1}_{[a,b]}$:

$$\int_{\mathbb{R}} k_{\mathrm{SE}, \sigma_1, \ell_1}(x_1, x_r)k_{\mathrm{SE}, \sigma_2, \ell_2}(x_r, x_2)\,\mathrm{d}\mu(x_r)$$
$$= \int_a^b c \cdot \sigma_1^2 \exp\left(-\frac{1}{2}\frac{(x_1 - x_r)^2}{\ell_1^2}\right) \cdot \sigma_2^2 \exp\left(-\frac{1}{2}\frac{(x_2 - x_r)^2}{\ell_2^2}\right)\,\mathrm{d}\lambda(x_r)$$
$$= c\sigma_1^2\sigma_2^2 \cdot \int_a^b \exp\left(-\frac{1}{2}\frac{(x_1 - x_r)^2}{\ell_1^2}\right) \cdot \exp\left(-\frac{1}{2}\frac{(x_2 - x_r)^2}{\ell_2^2}\right)\,\mathrm{d}\lambda(x_r)$$
$$= c\sigma_1^2\sigma_2^2 \cdot \int_a^b \exp\left(-\frac{\ell_2^2(x_1 - x_r)^2 - \ell_1^2(x_2 - x_r)^2}{2\ell_1^2\ell_2^2}\right)\,\mathrm{d}\lambda(x_r)$$
$$= c\sigma_1^2\sigma_2^2 \cdot \int_a^b \exp\left(-(\ell_1^2 + \ell_2^2)\cdot\left(x_r - \frac{2\ell_1^2 x_2 + 2\ell_2^2 x_1}{2(\ell_1^2 + \ell_2^2)}\right)^2 - \frac{\ell_1^2\ell_2^2(x_1 - x_2)^2}{\ell_1^2 + \ell_2^2}\right)\,\mathrm{d}\lambda(x_r)$$
$$= c\sigma_1^2\sigma_2^2 \cdot \frac{\sqrt{\pi}\ell_1\ell_2 \cdot \exp\left(-\frac{(x_1 - x_2)^2}{2(\ell_1^2 + \ell_2^2)}\right)\cdot\left(\mathrm{erf}\left(\frac{\ell_1^2(b - x_1) + \ell_2^2(b - x_2)}{\ell_1\ell_2\sqrt{2\ell_1^2 + 2\ell_2^2}}\right) - \mathrm{erf}\left(\frac{\ell_1^2(a - x_1) + \ell_2^2(a - x_2)}{\ell_1\ell_2\sqrt{2\ell_1^2 + 2\ell_2^2}}\right)\right)}{\sqrt{2\ell_1^2 + 2\ell_2^2}}$$

The same computation for a Gaussian measure $\mu$ follows from an additional completion to the square and the multidimensional case follows from the one-dimensional case and Fubini's theorem on the order of integration. We refer to Appendix C for more details on this covariance function for T-IMSPE and IMSPE.

**Example A.5.** Using the Risch-Algorithm as implemented in Maple, polynomial covariance functions, half-integer Matérn covariance functions, and Wiener process covariance are PyTorch-elementary covariance marginalizable w.r.t. both continuous uniform distribution on an interval or any Gaussian distribution. Additionally, these three covariance functions, together with constant and squared exponential covariance functions, are all pairwise PyTorch-elementary cross-covariance marginalizable.

Table 1: These tables shows pairs of standard covariance functions and whether they are PyTorch-elementary cross-covariance marginalizable w.r.t. any continuous uniform distribution on a multidimensional interval (left table) or any Gaussian distribution (right table). The reasons for this summary are given in the examples A.4, A.5, A.6, and A.7. The negative results stemm from the fact, that Risch's algorithm in Maple failed to compute the necessary integrals in Maple in closed form. A checkmark on the diagonal means that a covariance function is PyTorch-elementary covariance marginalizable. For the negative results in brackets for the cosine covariance function, see the comment in Example A.7. For the extension from cosine covariance functions to random Fourier features (RFF) see Example A.11.

| uniform distribution | polynomial | SE | Wiener | cosine | RFF | Matérn | RQ | periodic |
|---|---|---|---|---|---|---|---|---|
| polynomial | ✓ | ✓ | ✓ | ✓ | ✓ | ✓ | - | - |
| SE | ✓ | ✓ | ✓ | (-) | (-) | ✓ | - | - |
| Wiener process | ✓ | ✓ | ✓ | ✓ | ✓ | ✓ | - | - |
| cosine | ✓ | (-) | ✓ | ✓ | ✓ | ✓ | - | - |
| RFF | ✓ | (-) | ✓ | ✓ | ✓ | ✓ | - | - |
| Matérn | ✓ | ✓ | ✓ | ✓ | ✓ | ✓ | - | - |
| RQ | - | - | - | - | - | - | - | - |
| Periodic | - | - | - | - | - | - | - | - |

| Gaussian distribution | polynomial | SE | Wiener | cosine | RFF | Matérn | RQ | periodic |
|---|---|---|---|---|---|---|---|---|
| polynomial | ✓ | ✓ | ✓ | ✓ | ✓ | ✓ | - | - |
| SE | ✓ | ✓ | ✓ | ✓ | ✓ | ✓ | - | - |
| Wiener process | ✓ | ✓ | ✓ | (-) | (-) | ✓ | - | - |
| cosine | ✓ | ✓ | (-) | ✓ | ✓ | (-) | - | - |
| RFF | ✓ | ✓ | (-) | ✓ | ✓ | (-) | - | - |
| Matérn | ✓ | ✓ | ✓ | (-) | (-) | ✓ | - | - |
| RQ | - | - | - | - | - | - | - | - |
| Periodic | - | - | - | - | - | - | - | - |

The proof of this is easy to understand on a high level, although a detailed technical proof ends in a gory fight against indices, case distinctions, and factors[4]. First, using Fubini's theorem, everything reduces to the one-dimensional case. Second, case distinction in the minimum-function in the Wiener process covariance function is easily dealt with by splitting the one-dimensional integrals at its positions of non-differentiability; this results in integrals where only one case appears. Afterwards, the Wiener process is just a polynomial covariance. Now, the integrands are just products of polynomials and exponential functions with at most quadratic exponents. These integrals can be solved by—potentially ugly—combinations of partial integration, completion to squares in exponents, and usage of the Gaussian error function, similar to Example A.4.

**Example A.6.** Neither the rational quadratic (RQ) nor the periodic covariance function seem to be[5] PyTorch-elementary covariance marginalizable w.r.t. non-degenerate continuous uniform distributions on a multidimensional interval or non-degenerate Gaussian distributions, let alone PyTorch-elementary cross-covariance marginalizable w.r.t. any of the standard covariance functions.

**Example A.7.** Consider the PyTorch-elementary (cross-)covariance marginalizability of the cosine covariance function w.r.t. a uniform distribution on an interval. It is easily recognized as PyTorch-elementary covariance marginalizable, as the integral basically reduces to a square of cosines. Similarly, Risch's algorithm in Maple is able to verify that the cosine covariance function is PyTorch-elementary cross-covariance marginalizable together with constant, linear, polynomial, and Wiener process covariance functions. More interestingly, it seems that the pair of squared exponential and cosine covariance function is not PyTorch-elementary cross-covariance marginalizable w.r.t. both continuous uniform distribution on an interval or any Gaussian distribution. While Maple can find an antiderivative for the defining integral in closed form, this antiderivative involves complex error functions[6], which are not implemented in current versions of PyTorch. Perhaps, this pair will be PyTorch-elementary cross-covariance marginalizable in a future version of PyTorch.

---

[4]We recommend implementations of the Risch-Algorithm as implemented in Maple, Maxima, or Mathematica to solve these integrals.

[5]At least neither the authors nor—and much more important—the implementation of the Risch algorithm in Maple were able to compute these integrals in closed form.

[6]The following Python code shows, that the error function in PyTorch does not support complex arguments in a recent version.

```
>> torch.__version__
'2.0.1'
>> torch.erf(torch.tensor(complex(1,1)))
Traceback (most recent call last):
File "<stdin>", line 1, in <module>
RuntimeError:  "erf_vml_cpu" not implemented for 'ComplexFloat'
```

Now consider the cosine covariance function and its PyTorch-elementary (cross-)covariance marginalizable w.r.t. a Gaussian distribution. Even showing that it is PyTorch-elementary covariance marginalizable needs some manual intervention to simplify the results of Risch's algorithm in Maple[7]. As the squared exponential covariance function is a Gaussian, this trick works to show that the pair of cosine covariance function and squared exponential covariance function are PyTorch-elementary cross-covariance marginalizable. The pair of cosine covariance and polynomial covariance is easily seen as PyTorch-elementary cross-covariance marginalizable using repeated partial integration. Similar to above, we get complex error functions when trying to compute the integrals for the Pytorch-elementary cross-covariance marginalizability of the cosine covariance when paired with either half-integer Matérn covariance or Wieder process covariance, rendering these two pairs not Pytorch-elementary cross-covariance marginalizability in current version of PyTorch.

We summarize the results of the previous examples A.4, A.5, A.6, and A.7 in Table 1. We do not recommend to reproduce most of these integrals without the help of a computer algebra system.

### A.3   A cookbook of constructing $P$-elementary covariance marginalizable covariance functions

Knowing the base covariance functions, which are $P$-elementary (cross-)covariance marginalizable, we now consider the standard constructions of composite covariance functions. We construct a cookbook that states which of these constructions results in $P$-elementary (cross-)covariance marginalizable covariance functions. As a first example, scaling covariance functions does not change their $P$-elementary cross-covariance marginalizability.

**Proposition A.8** (Scaling covariance functions). *If a pair $(k_1, k_2)$ of two covariance functions is $P$-elementary cross-covariance marginalizable w.r.t. some measure $\mu$ then so is $(\sigma_1^2 k_1, \sigma_2^2 k_2)$ for any $\sigma_1, \sigma_2 > 0$.*

*Proof.* The proof is obvious from the linearity of integrals. □

Multiplying covariance functions on independent inputs keeps $P$-elementary cross-covariance marginalizability.

**Proposition A.9** (Covariance functions with independent inputs). *Consider a direct sum decomposition $\mathbb{R}^d \cong \mathbb{R}^{d_1} \oplus \mathbb{R}^{d_2}$ with $\mu_1$ a finite measure on $\mathbb{R}^{d_1}$ and $\mu_2$ a finite measure on $\mathbb{R}^{d_2}$. Assume that a covariance function $k$ factors over this decomposition, i.e. there are two covariance functions $k_i : \mathbb{R}^{d_i} \oplus \mathbb{R}^{d_i} \to \mathbb{R}$, $i = 1, 2$, with $k(x, x') = k_1(x_1, x_1') \cdot k_2(x_2, x_2')$ for $x = (x_1, x_2) \in \mathbb{R}^{d_1} \oplus \mathbb{R}^{d_2}$ and $x' = (x_1', x_2') \in \mathbb{R}^{d_1} \oplus \mathbb{R}^{d_2}$. If $k_1$ resp. $k_2$ are $P$-elementary cross-covariance marginalizable w.r.t. $\mu_1$ resp. $\mu_2$, then so is $k$ w.r.t. $\mu_1 \otimes \mu_2$.*

*Proof.* This follows directly from Fubini's theorem on the order of integration. □

One special case of this previous result is that the measure $\mu$ might be chosen as a product measure of a bounded uniform measure in some coordinate directions and a Gaussian measure in different coordinate directions.

The class of $P$-elementary covariance marginalizable covariance functions is not closed under addition, for example the sum of a cosine covariance function and various other covariance functions is not Pytorch-elementary covariance marginalizable w.r.t. uniform or Gaussian measures. However, the addition of pairs of $P$-elementary covariance marginalizable covariance functions sometimes results in $P$-elementary covariance marginalizable covariance functions.

**Proposition A.10** (Sums of covariance functions). *Assume that a pair $(k_1, k_2)$ of two covariance functions is $P$-elementary cross-covariance marginalizable and both $k_1$ and $k_2$ are $P$-elementary covariance marginalizable w.r.t. some measure $\mu$. Then, $k_1 + k_2$ is $P$-elementary covariance marginalizable.*

---

[7]For simplicity we consider a special case where parameters are set to 1. The following Maple code computes of the relevant integral for the PyTorch-elementary covariance marginalizability of the cosine covariance function, subtracts the intended results, and shows that this difference is zero. A direct computation seems impossible.

```
> simplify(
      int(cos(x-xr)*cos(y-xr)*exp(-xr^2),xr=-infinity..infinity)
      -sqrt(Pi)/2*cos(x-y)-cos(x+y)*sqrt(Pi)/2*exp(-1));
                                     0
```

*Proof.* Consider

$$\int (k_1(x_1, x_r) + k_2(x_1, x_r))(k_1(x_2, x_r) + k_2(x_2, x_r)) \, \mathrm{d}\mu(x_r)$$

$$= \int k_1(x_1, x_r)k_1(x_2, x_r) + k_1(x_1, x_r)k_2(x_2, x_r)$$

$$+ k_2(x_1, x_r)k_1(x_2, x_r) + k_2(x_1, x_r)k_2(x_2, x_r) \, \mathrm{d}\mu(x_r)$$

$$= \int k_1(x_1, x_r)k_1(x_2, x_r) \, \mathrm{d}\mu(x_r) + \int k_1(x_1, x_r)k_2(x_2, x_r) \, \mathrm{d}\mu(x_r)$$

$$+ \int k_2(x_1, x_r)k_1(x_2, x_r) \, \mathrm{d}\mu(x_r) + \int k_2(x_1, x_r)k_2(x_2, x_r) \, \mathrm{d}\mu(x_r)$$

All four integrals exist as $P$-elementary functions by assumption. Since the class of $P$-elementary functions is closed under addition, the proposition holds. □

**Example A.11.** By this proposition, all said for the cosine covariance function in Example A.7 also holds for covariance functions from random Fourier features (Hensman et al., 2018), as these are just specific linear combinations of cosine covariance functions.

As a direct corollary, if three or more covariance functions are all $P$-elementary covariance marginalizable and pairwise $P$-elementary cross-covariance marginalizable w.r.t. some finite measure $\mu$, then so is their sum. Looking at Table 1 we hence see that e.g. the sum of a polynomial covariance function, a Wiener process, and a squared exponential covariance function is $P$-elementary covariance marginalizable.

Adding measures keeps $P$-elementary cross-covariance marginalizability.

**Proposition A.12** (Linear combinations of measures)**.** *If a pair $(k_1, k_2)$ of two covariance functions is $P$-elementary cross-covariance marginalizable w.r.t. both the measures $\mu_1$ and $\mu_2$, then so is $(k_1, k_2)$ w.r.t. $a_1\mu_1 + a_2\mu_2$ for $a_1, a_2 \in \mathbb{R}$.*

*Proof.* The proof is obvious from the linearity of integrals. □

This fact allows to use IMSPE and T-IMSPE in domains that are no hyperrectangles, as long as the domain can be reasonably approximated by hyperrectangles. We can even subtract measures this way, e.g. to cut out parts of an area. One could even use negative measures as obtained from this proposition to avoid getting information in certain areas.

## B Proofs of closed form computability of IMSPE and T-IMSPE

We provide our proof of Theorem 4.4 about the computability of IMSPE and T-IMSPE for $P$-elementary covariance marginalizable covariance functions and its corollaries. We do not claim novelty to the idea of this proof, since similar proofs exist for the special case of polynomial covariance functions since (Sacks et al., 1989b). The discussion of additional special cases are spread over the subsequent literature, see Subsection 3.1, e.g. Lemma 3.1 in (Binois et al., 2019). In comparison to previous proofs, we provide a longer and more explicit proof.

*Proof of Theorem 4.4.* Consider the $1 \times 1$-matrix $k(x_r | x_*, x)$ in the integral

$$\int k(x_r | x_*, x) \, \mathrm{d}\mu(x_r).$$

from the definition of IMSPE. We can make the integrand more explicit as

$$k(x_r | x, x_*) = k(x_r, x_r) - k(x_r, (x, x_*))k((x, x_*), (x, x_*))^{-1}k((x, x_*), x_r)$$

$$= k(x_r, x_r) - \begin{bmatrix} k(x_r, x) & k(x_r, x_*) \end{bmatrix} \begin{bmatrix} k(x, x) & k(x, x_*) \\ k(x_*, x) & k(x_*, x_*) \end{bmatrix}^{-1} \begin{bmatrix} k(x, x_r) \\ k(x_*, x_r) \end{bmatrix}$$

The Schur complement

$$S_* = k(x_*, x_*) - k(x_*, x)k(x, x)^{-1}k(x, x_*),$$

the scalar $\sigma_*^2 = k(x_*, x_*)$, and the vector

$$L := k(x_*, x)k(x, x)^{-1} = (k(x, x)^{-1}k(x, x_*))^T.$$

allows to describe the above matrix inverse in closed form.

$$
\begin{bmatrix} k(x, x) & k(x, x_*) \\ k(x_*, x) & k(x_*, x_*) \end{bmatrix}^{-1}
$$
$$
= \begin{bmatrix} k(x, x)^{-1} + k(x, x)^{-1}k(x, x_*)S_*^{-1}k(x_*, x)k(x, x)^{-1} & -k(x, x)^{-1}k(x, x_*)S_*^{-1} \\ -S_*^{-1}k(x_*, x)k(x, x)^{-1} & S_*^{-1} \end{bmatrix}
$$
$$
= \begin{bmatrix} k(x, x)^{-1} + S_*^{-1}L^T L & -S_*^{-1}L \\ -S_*^{-1}L & S_*^{-1} \end{bmatrix}
$$
$$
= S_*^{-1} \begin{bmatrix} S_* k(x, x)^{-1} + L^T L & -L^T \\ -L & 1 \end{bmatrix}
$$

Having a closed form of the inverse, we can write

$$k(x_r | x, x_*)$$
$$= k(x_r, x_r) - \begin{bmatrix} k(x_r, x) & k(x_r, x_*) \end{bmatrix} \begin{bmatrix} k(x, x) & k(x, x_*) \\ k(x_*, x) & k(x_*, x_*) \end{bmatrix}^{-1} \begin{bmatrix} k(x, x_r) \\ k(x_*, x_r) \end{bmatrix}$$
$$= k(x_r, x_r) - \begin{bmatrix} k(x_r, x) & k(x_r, x_*) \end{bmatrix} S_*^{-1} \begin{bmatrix} S_* k(x, x)^{-1} + L^T L & -L^T \\ -L & 1 \end{bmatrix} \begin{bmatrix} k(x, x_r) \\ k(x_*, x_r) \end{bmatrix}$$
$$= \sigma_*^2 - S_*^{-1} \begin{bmatrix} k(x_r, x) & k(x_r, x_*) \end{bmatrix} \begin{bmatrix} S_* k(x, x)^{-1} + L^T L & -L^T \\ -L & 1 \end{bmatrix} \begin{bmatrix} k(x, x_r) \\ k(x_*, x_r) \end{bmatrix}$$
$$= \sigma_*^2 - S_*^{-1} \begin{bmatrix} k(x_r, x) & k(x_r, x_*) \end{bmatrix} \begin{bmatrix} S_* k(x, x)^{-1}k(x, x_r) + L^T L k(x, x_r) - L^T k(x_*, x_r) \\ -L k(x, x_r) k(x_*, x_r) \end{bmatrix}$$
$$= \sigma_*^2 - S_*^{-1} \big( S_* k(x_r, x)k(x, x)^{-1}k(x, x_r) + k(x_r, x)L^T L k(x, x_r)$$
$$\qquad - k(x_r, x)L^T k(x_*, x_r) - k(x_r, x_*)L k(x, x_r) + k(x_r, x_*)k(x_*, x_r) \big)$$
$$= \sigma_*^2 - S_*^{-1} \big( S_* k(x_r, x)k(x, x)^{-1}k(x, x_r)$$
$$\qquad + k(x_r, x)k(x, x)^{-1}k(x, x_*)k(x_*, x)k(x, x)^{-1}k(x, x_r)$$
$$\qquad - k(x_r, x)k(x, x)^{-1}k(x, x_*)k(x_*, x_r)$$
$$\qquad - k(x_r, x_*)k(x_*, x)k(x, x)^{-1}k(x, x_r) + k(x_r, x_*)k(x_*, x_r) \big)$$

Writing $k(x, x) = C^T C$ as Cholesky decomposition, $C_v^T := k(x_r, x)/C^T$, $C_v := C \backslash k(x, x_r)$, $C_*^T := k(x_*, x)/C^T$, $C_* := C \backslash k(x, x_*)$, and the correlation $c = k(x_r, x_*) = k(x_*, x_r)$ this simplies to

$$k(x_r | x, x_*)$$
$$= \sigma_*^2 - S_*^{-1} \big( S_* k(x_r, x)/C^T C \backslash k(x, x_r) + k(x_r, x)/C^T C \backslash k(x, x_*)k(x_*, x)/C^T C \backslash k(x, x_r)$$
$$\qquad - k(x_r, x)/C^T C \backslash k(x, x_*)k(x_*, x_r) - k(x_r, x_*)k(x_*, x)/C^T C \backslash k(x, x_r)$$
$$\qquad + k(x_r, x_*)k(x_*, x_r) \big)$$
$$= \sigma_*^2 - S_*^{-1} \big( S_* C_v^T C_v + C_v^T C_* C_*^T C_v - c C_v^T C_* - c C_*^T C_v + c^2 \big)$$
$$= \sigma_*^2 - C_v^T C_v - S_*^{-1} C_v^T C_* C_*^T C_v + S_*^{-1} c C_v^T C_* + S_*^{-1} c C_*^T C_v - S_*^{-1} c^2$$

Now

$$\int k(x_r|x_*, x)\, \mathrm{d}\mu(x_r) \tag{I}$$

$$= \underbrace{\int \sigma_*^2\, \mathrm{d}\mu(x_r)}_{(I.1)} - \underbrace{\int C_v^T C_v\, \mathrm{d}\mu(x_r)}_{(I.2)} - \underbrace{\int S_*^{-1} C_v^T C_* C_*^T C_v\, \mathrm{d}\mu(x_r)}_{(I.3)}$$

$$+ \underbrace{\int S_*^{-1} c C_v^T C_*\, \mathrm{d}\mu(x_r)}_{(I.4)} + \underbrace{\int S_*^{-1} c C_*^T C_v\, \mathrm{d}\mu(x_r)}_{(I.5)} - \underbrace{\int S_*^{-1} c^2\, \mathrm{d}\mu(x_r)}_{(I.6)}$$

Here, $c^2$ and $S_*^{-1}$ are scalars, $C_v$ is a vector of linear combinations of $k(x_r, x)$, $C_*$ is a vector of linear combinations of $k(x_*, x)$. This means that all entries are either constant or a constant times a product of $k(x_r, z_1) \cdot k(x_r, z_2)$. All linear combinations and all constants can be computed via numerical linear algebra, mostly using the Cholesky decomposition. In particular, these integrals, considered as functions in $x_*$, are $P$-elementary, since $k$ is $P$-elementary covariance marginalizable.

T-IMSPE is a way of applying IMSPE to specific GPs. Hence, the claim for T-IMSPE follows from that of IMSPE. $\qquad\square$

*Proof of Corollary 4.5.* Initially, we compute a $O(n^3)$ Cholesky decomposition of the data covariance matrix $k(x, x)$. For each $x_*$, We make $O(n^2)$ evaluations of closed form functions and a small finite number of forward and backward substitutions of Cholesky factors, each computable in $O(n^2)$. $\qquad\square$

The statistics literature, see e.g. (Gramacy, 2020), is also concerned with avoiding the Cholesky decomposition in $O(n^3)$, which is possible by rank-one-update formulas of the covariance matrices. These computational improvements are only possible when hyperparameters are not retrained between iterations.

## C  Detailed explicit formulas of T-IMSPE and IMSPE for squared exponential covariance functions

We give very explicit formulas for T-IMSPE and IMPSE when using squared exponential covariance function. Therefore, consider the 6 integrals in (I) on page 27 independently. We assume a squared exponential covariance function

$$k : \mathbb{R}^d \times \mathbb{R}^d \to \mathbb{R} : (x_1, x_2) \mapsto \prod_{h=1}^d \sigma^2 \exp\left(-\frac{1}{2}\frac{((x_1)_h - (x_2)_h)^2}{\ell_h^2}\right)$$

$$= \sigma^2 \exp\left(-\frac{1}{2}\sum_{h=1}^d \frac{((x_1)_h - (x_2)_h)^2}{\ell_h^2}\right)$$

with automatic relevance determination. For simplicity, we assume $\mu$ to be a probability distribution, i.e. $\mu(\mathbb{R}^d) = 1$, and later we will also specifically concentrate on Gaussian measures and uniform measures.

### C.1  Integral (I.1)

$\int \sigma_*^2\, \mathrm{d}\mu(x_r) = \sigma_*^2 = k(x_*, x_*)$. This term is independent of $x_*$, as $k$ is stationary.

## C.2 Integral (I.2)

$$\int -C_v^T C_v \, \mathrm{d}\mu(x_r)$$

$$= -\int k(x_r, x)/C^T C \backslash k(x, x_r) \, \mathrm{d}\mu(x_r)$$

$$= -\int k(x_r, x) k(x, x)^{-1} k(x, x_r) \, \mathrm{d}\mu(x_r)$$

$$= -\sum_{i,j=1}^{n} \int k(x_r, x_i) \left(k(x, x)^{-1}\right)_{i,j} k(x_j, x_r) \, \mathrm{d}\mu(x_r)$$

$$= -\sum_{i,j=1}^{n} \left(k(x, x)^{-1}\right)_{i,j} \int k(x_r, x_i) k(x_j, x_r) \, \mathrm{d}\mu(x_r)$$

$$= -\sum_{i,j=1}^{n} \left(k(x, x)^{-1}\right)_{i,j} \cdot \sigma^4 \cdot \int \prod_{h=1}^{d} \exp\left(-\frac{1}{2} \frac{((x_r)_h - (x_i)_h)^2 + ((x_j)_h - (x_r)_h)^2}{\ell_h^2}\right) \mathrm{d}\mu(x_r)$$

$$= -\sum_{i,j=1}^{n} \left(k(x, x)^{-1}\right)_{i,j} \cdot \sigma^4 \cdot \prod_{h=1}^{d} \int \exp\left(-\frac{1}{2} \frac{((x_r)_h - (x_i)_h)^2 + ((x_j)_h - (x_r)_h)^2}{\ell_h^2}\right) \mathrm{d}\mu(x_r)$$

This should be integrable easily, e.g. for $\mu$ the continuous uniform distribution on $[a_1, b_1] \times \ldots \times [a_d, b_d]$ this integral evaluates to

$$-\sum_{i,j=1}^{n} \left(k(x, x)^{-1}\right)_{i,j} \cdot \sigma^4 \cdot \sqrt{\pi}^d$$

$$\cdot \prod_{h=1}^{d} \ell_h \cdot \exp\left(-\frac{((x_j)_h - (x_i)_h)^2}{4\ell_h^2}\right) \cdot \frac{\mathrm{erf}\left(\frac{2a_h - (x_j)_h - (x_i)_h}{2\ell_h}\right) - \mathrm{erf}\left(\frac{2b_h - (x_j)_h - (x_i)_h}{2\ell_h}\right)}{2a_h - 2b_h}$$

or for $\mu = \mathcal{N}(m, \mathrm{diag}(s))$ this integral evaluates to

$$-\sum_{i,j=1}^{n} \left(k(x, x)^{-1}\right)_{i,j} \cdot \sigma^4 \cdot \prod_{h=1}^{d} \ell_h \cdot \frac{\exp\left(\frac{-\ell_h^2((x_j)_h - m_h)^2 - \ell_h^2((x_i)_h - m_h)^2 - s_h^2((x_j)_h - (x_i)_h)^2}{2\ell_h^2(\ell_h^2 + 2s_h^2)}\right)}{\sqrt{\ell_h^2 + 2s_h^2}} \cdot$$

This term is independent of $x_*$.

## C.3 Integral (I.3)

$$\int -S_*^{-1} C_v^T C_* C_*^T C_v \, \mathrm{d}\mu(x_r)$$

$$= -S_*^{-1} \int C_v^T C_* C_*^T C_v \, \mathrm{d}\mu(x_r)$$

$$= -S_*^{-1} \int k(x_r, x)/C^T C\backslash k(x, x_*) k(x_*, x)/C^T C\backslash k(x, x_r) \mu(x_r)$$

$$= -S_*^{-1} \int k(x_r, x) \cdot (C^T\backslash C\backslash k(x, x_*))(k(x_*, x)/C^T/C) \cdot k(x, x_r) \mu(x_r)$$

$$= -S_*^{-1} \int k(x_r, x) \cdot (\kappa \kappa^T) \cdot k(x, x_r) \mu(x_r)$$

$$= -S_*^{-1} \sum_{i,j=1}^{n} \int k(x_r, x_i) \cdot \kappa_i \kappa_j \cdot k(x_j, x_r) \mu(x_r)$$

$$= -S_*^{-1} \sum_{i,j=1}^{n} \kappa_i \kappa_j \int k(x_r, x_i) \cdot k(x_j, x_r) \mu(x_r)$$

$$= -S_*^{-1} \sigma^4 \sum_{i,j=1}^{n} \kappa_i \kappa_j \int \prod_{h=1}^{d} \exp\left( -\frac{1}{2} \frac{(x_r - x_i)^2 + (x_j - x_r)^2}{\ell^2} \right) \mu(x_r)$$

$$= -S_*^{-1} \sigma^4 \sum_{i,j=1}^{n} \kappa_i \kappa_j \prod_{h=1}^{d} \int \exp\left( -\frac{1}{2} \frac{(x_r - x_i)^2 + (x_j - x_r)^2}{\ell^2} \right) \mu(x_r)$$

for $\kappa = \left( C^T\backslash C\backslash k(x, x_*) \right) = k(x, x)^{-1} k(x, x_*)$. This should be integrable easily, e.g. for $\mu$ the continuous uniform distribution on $[a_1, b_1] \times \ldots \times [a_d, b_d]$ this integral evaluates to

$$- \sum_{i,j=1}^{n} S_*^{-1} \sigma^4 \kappa_i \kappa_j \sqrt{\pi}^d \cdot \prod_{h=1}^{d} \ell_h$$

$$\cdot \prod_{h=1}^{d} \exp\left( -\frac{((x_j)_h - (x_i)_h)^2}{4\ell_h^2} \right) \cdot \frac{\mathrm{erf}\left( \frac{2a_h - (x_j)_h - (x_i)_h}{2\ell_h} \right) - \mathrm{erf}\left( \frac{2b_h - (x_j)_h - (x_i)_h}{2\ell_h} \right)}{2a_h - 2b_h}$$

or for $\mu = \mathcal{N}(m, \mathrm{diag}(s))$ this integral evaluates to

$$- \sum_{i,j=1}^{n} S_*^{-1} \sigma^4 \kappa_i \kappa_j \cdot \prod_{h=1}^{d} \ell_h \cdot \frac{\exp\left( \frac{-\ell_h^2 ((x_j)_h - m_h)^2 - \ell_h^2 ((x_i)_h - m_h)^2 - s_h^2 ((x_j)_h - (x_i)_h)^2}{2\ell_h^2 (\ell_h^2 + 2s_h^2)} \right)}{\sqrt{\ell_h^2 + 2s_h^2}}.$$

This term depends on $x_*$ (via $\kappa$).

### C.4 Integral (I.4)

$$\int S_*^{-1} c C_v^T C_* \mu(x_r)$$

$$= \int S_*^{-1} k(x_*, x_r) k(x_r, x) / C^T C \backslash k(x, x_*) \mu(x_r)$$

$$= S_*^{-1} \sigma^2 \int \prod_{h=1}^{d} \exp\left(-\frac{1}{2}\frac{((x_*)_h - (x_r)_h)^2}{\ell_h^2}\right) k(x_r, x) / C^T C \backslash k(x, x_*) \mu(x_r)$$

$$= S_*^{-1} \sigma^2 \int \prod_{h=1}^{d} \exp\left(-\frac{1}{2}\frac{((x_*)_h - (x_r)_h)^2}{\ell_h^2}\right) k(x_r, x) \cdot C^T \backslash C \backslash k(x, x_*) \mu(x_r)$$

$$= S_*^{-1} \sigma^2 \sum_{i=1}^{n} \kappa_i \cdot \int \prod_{h=1}^{d} \exp\left(-\frac{1}{2}\frac{((x_*)_h - (x_r)_h)^2}{\ell_h^2}\right) k(x_r, x_i) \mu(x_r)$$

$$= \sum_{i=1}^{n} \kappa_i \cdot S_*^{-1} \sigma^4 \cdot \int \prod_{h=1}^{d} \exp\left(-\frac{1}{2}\frac{((x_*)_h - (x_r)_h)^2 + ((x_r)_h - (x_i)_h)^2}{\ell_h^2}\right) \mu(x_r)$$

$$= \sum_{i=1}^{n} \kappa_i \cdot S_*^{-1} \sigma^4 \cdot \prod_{h=1}^{d} \int \exp\left(-\frac{1}{2}\frac{((x_*)_h - (x_r)_h)^2 + ((x_r)_h - (x_i)_h)^2}{\ell_h^2}\right) \mu(x_r)$$

for $\kappa = \left(C^T \backslash C \backslash k(x, x_*)\right) = k(x, x)^{-1} k(x, x_*)$. This should be integrable easily, e.g. for $\mu$ the continuous uniform distribution on $[a_1, b_1] \times \ldots \times [a_d, b_d]$ this integral evaluates to

$$\sum_{i=1}^{n} \kappa_i \cdot S_*^{-1} \sigma^4 \cdot \sqrt{\pi}^d \cdot \prod_{h=1}^{d} \ell_h$$

$$\cdot \prod_{h=1}^{d} \exp\left(-\frac{((x_*)_h - (x_i)_h)^2}{4\ell_h^2}\right) \cdot \frac{\operatorname{erf}\left(\frac{2a_h - (x_*)_h - (x_i)_h}{2\ell_h}\right) - \operatorname{erf}\left(\frac{2b_h - (x_*)_h - (x_i)_h}{2\ell_h}\right)}{2a_h - 2b_h}$$

or for $\mu = \mathcal{N}(m, \operatorname{diag}(s))$ this integral evaluates to

$$\sum_{i=1}^{n} \kappa_i \cdot S_*^{-1} \sigma^4 \cdot \prod_{h=1}^{d} \ell_h \cdot \frac{\exp\left(\frac{-\ell_h^2((x_*)_h - m_h)^2 - \ell_h^2((x_i)_h - m_h)^2 - s_h^2((x_*)_h - (x_i)_h)^2}{2\ell_h^2(\ell_h^2 + 2s_h^2)}\right)}{\sqrt{\ell_h^2 + 2s_h^2}}.$$

This term depends on $x_*$.

### C.5 Integral (I.5)

$\int S_*^{-1} c C_*^T C_v \mu(x_r)$. This is the same integral as the last one.

### C.6 Integral (I.6)

$$\int -S_*^{-1} c^2 \, d\mu(x_r) = \int -S_*^{-1} k(x_r, x_*)^2 \, d\mu(x_r)$$

$$= \int -S_*^{-1} \sigma^4 \prod_{h=1}^{d} \exp\left(-\frac{1}{2}\frac{((x_r)_h - (x_*)_h)^2}{\ell_h^2}\right)^2 \, d\mu(x_r)$$

$$= -S_*^{-1} \sigma^4 \prod_{h=1}^{d} \int \exp\left(-\frac{((x_r)_h - (x_*)_h)^2}{\ell_h^2}\right)^2 \, d\mu(x_r)$$

This is easily integrable, e.g. for $\mu$ the continuous uniform distribution on $[a_1, b_1] \times \ldots \times [a_d, b_d]$ this integral evaluates to

$$-S_*^{-1}\sigma^4 \cdot \sqrt{\pi}^d \cdot \prod_{h=1}^{d} \ell_h \cdot \frac{\operatorname{erf}(\frac{a_h - (x_*)_h}{\ell_h}) - \operatorname{erf}(\frac{b_h - (x_*)_h}{\ell_h})}{2a_h - 2b_h}$$

or for $\mu = \mathcal{N}(m, \operatorname{diag}(s))$ this integral evaluates to

$$-S_*^{-1}\sigma^4 \cdot \prod_{h=1}^{d} \frac{\ell_h \cdot \exp(-\frac{(m_h - (x_*)_h)^2}{\ell_h^2 + 2s_h^2})}{\sqrt{\ell_h^2 + 2s_h^2}}.$$

This term depends on $x_*$.

### C.7 All terms in (I) together

For $\mu$ the continuous uniform distribution on $[a, b]$ we have:

$$\int k(x_r | x_*, x) \, d\mu(x_r)$$

$$= \sigma_*^2$$

$$- \sum_{i,j=1}^{n} \left(k(x,x)^{-1}\right)_{i,j} \cdot \sigma^4 \cdot \sqrt{\pi}^d \cdot \prod_{h=1}^{d} \ell_h$$

$$\cdot \prod_{h=1}^{d} \exp\left(-\frac{((x_j)_h - (x_i)_h)^2}{4\ell_h^2}\right) \cdot \frac{\operatorname{erf}\left(\frac{2a_h - (x_j)_h - (x_i)_h}{2\ell_h}\right) - \operatorname{erf}\left(\frac{2b_h - (x_j)_h - (x_i)_h}{2\ell_h}\right)}{2a_h - 2b_h}$$

$$- \sum_{i,j=1}^{n} S_*^{-1}\sigma^4 \kappa_i \kappa_j \sqrt{\pi}^d \cdot \prod_{h=1}^{d} \ell_h$$

$$\cdot \prod_{h=1}^{d} \exp\left(-\frac{((x_j)_h - (x_i)_h)^2}{4\ell_h^2}\right) \cdot \frac{\operatorname{erf}\left(\frac{2a_h - (x_j)_h - (x_i)_h}{2\ell_h}\right) - \operatorname{erf}\left(\frac{2b_h - (x_j)_h - (x_i)_h}{2\ell_h}\right)}{2a_h - 2b_h}$$

$$+ 2\sum_{i=1}^{n} \kappa_i \cdot S_*^{-1}\sigma^4 \cdot \sqrt{\pi}^d \cdot \prod_{h=1}^{d} \ell_h$$

$$\cdot \prod_{h=1}^{d} \exp\left(-\frac{((x_*)_h - (x_i)_h)^2}{4\ell_h^2}\right) \cdot \frac{\operatorname{erf}\left(\frac{2a_h - (x_*)_h - (x_i)_h}{2\ell_h}\right) - \operatorname{erf}\left(\frac{2b_h - (x_*)_h - (x_i)_h}{2\ell_h}\right)}{2a_h - 2b_h}$$

$$- S_*^{-1}\sigma^4 \cdot \sqrt{\pi}^d \cdot \prod_{h=1}^{d} \ell_h \cdot \frac{\operatorname{erf}(\frac{a_h - (x_*)_h}{\ell_h}) - \operatorname{erf}(\frac{b_h - (x_*)_h}{\ell_h})}{2a_h - 2b_h}$$

$$= \sigma_*^2$$

$$- \sum_{i,j=1}^{n} \left(\left(k(x,x)^{-1}\right)_{i,j} + S_*^{-1}\kappa_i \kappa_j\right) \cdot \sigma^4 \cdot \sqrt{\pi}^d \cdot \prod_{h=1}^{d} \ell_h$$

$$\cdot \prod_{h=1}^{d} \exp\left(-\frac{((x_j)_h - (x_i)_h)^2}{4\ell_h^2}\right) \cdot \frac{\operatorname{erf}\left(\frac{2a_h - (x_j)_h - (x_i)_h}{2\ell_h}\right) - \operatorname{erf}\left(\frac{2b_h - (x_j)_h - (x_i)_h}{2\ell_h}\right)}{2a_h - 2b_h}$$

$$+ 2\sum_{i=1}^{n} \kappa_i \cdot S_*^{-1}\sigma^4 \cdot \sqrt{\pi}^d \cdot \prod_{h=1}^{d} \ell_h$$

$$\cdot \prod_{h=1}^{d} \exp\left(-\frac{((x_*)_h - (x_i)_h)^2}{4\ell_h^2}\right) \cdot \frac{\mathrm{erf}\left(\frac{2a_h - (x_*)_h - (x_i)_h}{2\ell_h}\right) - \mathrm{erf}\left(\frac{2b_h - (x_*)_h - (x_i)_h}{2\ell_h}\right)}{2a_h - 2b_h}$$

$$- S_*^{-1}\sigma^4 \cdot \sqrt{\pi}^d \cdot \prod_{h=1}^{d} \ell_h \cdot \frac{\mathrm{erf}(\frac{a_h - (x_*)_h}{\ell_h}) - \mathrm{erf}(\frac{b_h - (x_*)_h}{\ell_h})}{2a_h - 2b_h}$$

For a multivariate normal $\mu = \mathcal{N}(m, s)$ where $s = \mathrm{diag}(s_1, \ldots, s_d) \in \mathbb{R}^{d \times d}$ we have:

$$\int k(x_r | x_*, x) \,\mathrm{d}\mu(x_r)$$

$$= \sigma_*^2$$

$$- \sum_{i,j=1}^{n} \left(k(x,x)^{-1}\right)_{i,j} \cdot \sigma^4 \cdot \prod_{h=1}^{d} \ell_h \cdot \frac{\exp\left(\frac{-\ell_h^2((x_j)_h - m_h)^2 - \ell_h^2((x_i)_h - m_h)^2 - s_h^2((x_j)_h - (x_i)_h)^2}{2\ell_h^2(\ell_h^2 + 2s_h^2)}\right)}{\sqrt{\ell_h^2 + 2s_h^2}}$$

$$- \sum_{i,j=1}^{n} S_*^{-1}\sigma^4 \kappa_i \kappa_j \cdot \prod_{h=1}^{d} \ell_h \cdot \frac{\exp\left(\frac{-\ell_h^2((x_j)_h - m_h)^2 - \ell_h^2((x_i)_h - m_h)^2 - s_h^2((x_j)_h - (x_i)_h)^2}{2\ell_h^2(\ell_h^2 + 2s_h^2)}\right)}{\sqrt{\ell_h^2 + 2s_h^2}}$$

$$+ 2\sum_{i=1}^{n} \kappa_i \cdot S_*^{-1}\sigma^4 \cdot \prod_{h=1}^{d} \ell_h \cdot \frac{\exp\left(\frac{-\ell_h^2((x_*)_h - m_h)^2 - \ell_h^2((x_i)_h - m_h)^2 - s_h^2((x_*)_h - (x_i)_h)^2}{2\ell_h^2(\ell_h^2 + 2s_h^2)}\right)}{\sqrt{\ell_h^2 + 2s_h^2}}$$

$$- S_*^{-1}\sigma^4 \cdot \prod_{h=1}^{d} \frac{\ell_h \cdot \exp\left(-\frac{(m_h - (x_*)_h)^2}{\ell_h^2 + 2s_h^2}\right)}{\sqrt{\ell_h^2 + 2s_h^2}}$$

$$= \sigma_*^2$$

$$- \sum_{i,j=1}^{n} \left(\left(k(x,x)^{-1}\right)_{i,j} + S_*^{-1}\kappa_i \kappa_j\right) \cdot \sigma^4 \cdot \prod_{h=1}^{d} \ell_h$$

$$\prod_{h=1}^{d} \cdot \frac{\exp\left(\frac{-\ell_h^2((x_j)_h - m_h)^2 - \ell_h^2((x_i)_h - m_h)^2 - s_h^2((x_j)_h - (x_i)_h)^2}{2\ell_h^2(\ell_h^2 + 2s_h^2)}\right)}{\sqrt{\ell_h^2 + 2s_h^2}}$$

$$+ 2\sum_{i=1}^{n} \kappa_i \cdot S_*^{-1}\sigma^4 \cdot \prod_{h=1}^{d} \ell_h \cdot \frac{\exp\left(\frac{-\ell_h^2((x_*)_h - m_h)^2 - \ell_h^2((x_i)_h - m_h)^2 - s_h^2((x_*)_h - (x_i)_h)^2}{2\ell_h^2(\ell_h^2 + 2s_h^2)}\right)}{\sqrt{\ell_h^2 + 2s_h^2}}$$

$$- S_*^{-1}\sigma^4 \cdot \prod_{h=1}^{d} \frac{\ell_h \cdot \exp\left(-\frac{(m_h - (x_*)_h)^2}{\ell_h^2 + 2s_h^2}\right)}{\sqrt{\ell_h^2 + 2s_h^2}}$$

## D  On Baselines

Here, we comment on our choice for entropy as the only baselines in the experimental Section 5. We chose entropy as (only) baseline acquisition function, since it is the state of the art for active learning and safe active learning in machine learning. This is different to Bayesian optimization, where multiple acquisition functions are commonly used. In Appendix D.1 we give reasons why several classical methods are no suitable baselines. Marginalized $\beta$-diversity and marginalized mutual information are acquisition functions we developed during the paper and do not appear in the literature. They have no closed form, and their numerical approximations are very sub-par in our preliminary ablation experiments in Appendix D.2. Furthermore, we comment the superiority of entropy over IMSPE for GPs with trainable hyperparameters as remarked in Subsection 2.4, which excludes IMSPE as baseline.

### D.1 Classical baselines

ALM is just another name for the entropy, as described in Section 2.3. D-optimal designs are a non-active version of entropy, as described in Section 3.1. IMSE, ALC, IV, and IVAR are just other names of IMSPE, as commented in Section 3.1. A-optimal designs and V-optimal designs are non-active versions of IMSPE in different interpretations, and hence not suitable for a comparison.

### D.2 Comparison of IMSPE to marginalizing $\beta$-diversity and marginalizing mutual information

Maximizing the mutual information $I(x_*; x_r \mid x)$ (Krause et al., 2008) between a new datapoint $x_*$ w.r.t. some points of interest $x_r$ conditioned on previously existing data $x$ is a classic choice to place sensors or to conduct measurements in active learning settings. It is originally restricted to discrete sensor positions, where it leads to an NP hard decision problem. Despite this restriction, applicability of mutual information has increased as discussed in Subsection 3.1.

For active learning purposes we can replace the mutual information by the squared $\beta$-diversity[8] $D_\beta^2(x_*; x_r \mid x) := \exp(I(x_*; x_r))$ (Leinster, 2021; van Dam, 2019; Chiu et al., 2014), as is is just a composition with a monotonous and injective map. Instead of maximizing the (squared) $\beta$-diversity, we can also minimize the inverse squared[9] $\beta$-diversity $D_\beta^{-2}(x_*; x_r | x)$. Consider $x_*, x_r \in \mathbb{R}^{1 \times d}$ and $x \in \mathbb{R}^{n \times d}$.

$$
\begin{aligned}
D_\beta^{-2}(x_*; x_r \mid x) &= \exp(I(x_*; x_r \mid x))^{-2} \\
&= \exp\left(H(x_r \mid x) - H(x_r \mid x_*, x)\right)^{-2} \\
&= \frac{\exp\left(\frac{d}{2}(1 + \log(2\pi)) + \frac{1}{2}\log\left(\det\left(k(x_r|x_*, x)\right)\right)\right)^2}{\exp\left(\left(\frac{d}{2}(1 + \log(2\pi)) + \frac{1}{2}\log\left(\det\left(k(x_r|x)\right)\right)\right)\right)^2} \\
&= \frac{\exp\left(\log\left(\det\left(k(x_r|x_*, x)\right)\right)\right)}{\exp\left(\log\left(\det\left(k(x_r|x)\right)\right)\right)} \\
&= \frac{\det\left(k(x_r|x_*, x)\right)}{\det\left(k(x_r|x)\right)} \\
&= \frac{k(x_r|x_*, x)}{k(x_r|x)} \qquad\qquad \text{determinant of 1x1-matrix}
\end{aligned}
$$

The denominator is a constant w.r.t. $x_*$ and can be disregarded when minimizing $D_\beta^{-2}(x_*; x_r | x)$ w.r.t. $x_*$. We can write the numerator in terms of prior covariances:

$$
k(x_r|x_*, x) = k(x_r, x_r) - k(x_r, (x_*, x))k((x_*, x), (x_*, x))^{-1}k((x_*, x), x_r)
$$

Minimizing this acquisition function obviously leads to choosing $x_* \approx x_r$, which is obviously a usually bad choice, since any real form ob optimization is ignored in favor of just choosing the reference point. Similar phenomena appear for multiple reference points $x_r$, where each reference point leads to a local minima in the loss landscape of $x_* \mapsto k(x_r|x_*, x)$. To prevent these attracting local minima in the loss landscape, one can minimize the above criteria when averaged over $x_r$:

$$
\int D_\beta^{-2}(x_*; x_r \mid x)\, d\mu(x_r) = \int \frac{k(x_r|x_*, x)}{k(x_r|x)}\, d\mu(x_r) \qquad \text{average inverse squared } \beta\text{-diversity}
$$

$$
\int k(x_r|x_*, x)\, d\mu(x_r) \qquad\qquad\qquad\qquad\quad \text{IMSPE}
$$

$$
\int -I(x_*; x_r \mid x)\, d\mu(x_r) \qquad\qquad\qquad\qquad \text{average negative mutual information}
$$

---

[8]The interpretation of $\beta$-diversity is that of the diversity between two groups. In our case, this we aim to minimize the diversity between $x_*$ and $x_r$, when conditioned on $x$.

[9]We were unable to find anything close to resembling a closed formula for any other power of the $\beta$-diversity.

Here, we assume some (finite or probability) measure $\mu$.

The average negative mutual information and the IMSPE are connected via a Jensen gap.

$$
\begin{aligned}
\int & - \mathrm{I}(x_*; x_r \mid x)\,\mathrm{d}\mu(x_r) \\
&= -\int \left( \frac{1}{2} \log\left(\det\left(k(x_r|x)\right)\right) - \frac{1}{2}\log\left(\det\left(k(x_r|x_*,x)\right)\right) \right) \mathrm{d}\mu(x_r) \\
&= -\int \left( \frac{1}{2} \log\left(k(x_r|x)\right) - \frac{1}{2}\log\left(k(x_r|x_*,x)\right) \right) \mathrm{d}\mu(x_r) \\
&= \frac{1}{2}\int \log\left(k(x_r|x_*,x)\right)\mathrm{d}\mu(x_r) - \frac{1}{2}\int \log\left(k(x_r|x)\right)\mathrm{d}\mu(x_r) \\
&\stackrel{\text{Jensen}}{\leq} \frac{1}{2}\log\left( \int k(x_r|x_*,x)\,\mathrm{d}\mu(x_r) \right) - \underbrace{\frac{1}{2}\int \log\left(k(x_r|x)\right)\mathrm{d}\mu(x_r)}_{\text{constant in } x_*} \\
&\stackrel{\pm}{=} \frac{1}{2}\log\left( \int k(x_r|x_*,x)\,\mathrm{d}\mu(x_r) \right) \quad \text{(equal up to a constant)}
\end{aligned}
$$

Here, the latter line is just one half the log of the acquisition function IMSPE. Hence, the mutual information and IMSPE are closely connected. IMSPE has the advantage for continuous domains, as one can marginalize the reference points in closed form.

The comparison to the inverse squared $\beta$-divergence $D_\beta^{-2}(x_*; x_r \mid x)$ is more complicated. While $D_\beta^{-2}(x_*; x_r \mid x) = \frac{k(x_r|x_*,x)}{k(x_r|x)}$ and $k(x_r|x_*,x)$ are equal up to a factor that is constant in $x_*$. Sadly, this factor is not constant in $x_r$, over which we marginalize. Hence, $\int k(x_r|x_*,x)\,\mathrm{d}\mu(x_r)$ and $\int D_\beta^{-2}(x_*; x_r \mid x)\,\mathrm{d}\mu(x_r) = \int \frac{k(x_r|x_*,x)}{k(x_r|x)}\,\mathrm{d}\mu(x_r)$ are differently weighed integrals. In the latter case, one weighs reference points $x_r$ higher if $k(x_r|x) \approx 0$, i.e. reference points near the data points. We do not think that this is suitable in practice. In addition to no closed form integrals and very suboptimal initial experiments, we did not pursue the $\beta$-diversity any further.

We elaborate on a preliminary empirical comparison between the marginalizations suggested above, and not only between (T-)IMSPE and entropy. First, we were neither able to compute $\int - \mathrm{I}(x_*; x_r \mid x)\,\mathrm{d}\mu(x_r)$ nor $\int D_\beta^{-2}(x_*; x_r \mid x)\,\mathrm{d}\mu(x_r)$ in closed form for any choice of covariance function we tried. Furthermore, numerical or MC approximations to these integrals suffer from a problem: the appearance of local optima at all points where the integral is numerically or stochastically evaluated. In preliminary experiments, this led to the same suboptimal behavior as for finitely many reference points $x_r$. More precisely, for the minimization of both $-I(x_*; x_r|x)$ and $D_\beta^{-2}(x_*; x_r \mid x)$, one chooses $x_*$ is a local optimum close to one of the $x_r$. In fact, any data point in $x$ will repel $x_*$ and any point in $x_r$ will attract $x_*$. This leads to many local optima, that are not optimal for active learning. The many local optima are a problem for the optimization in safe active learning, as many more starts of the optimization are necessary, leading to drastically increased computation time. Further increasing the number of reference points $x_r$ leads to more, but less pronounced, local extrema; this did not improve the situation. In Figure 7 we see how optimal the acquisition functions entropy, IMSPE, and a numerical approximation of IMSPE are for placing a single data points in one to twelve dimension.

We cannot change the $x_r$ during a single optimization run, as this leads to unstable optimization. Even choosing the $x_r$ different in every active learning step resulted in suboptimal exploration, as measurements are not prioritized by being near the safety boundary, but by being close to one of the $x_r$.

# E   Further results of the experiments

## E.1   Further results for the seasonal change experiment

For the example about seasonal change from Subsection 5.1 the average amount of safe area that is recognized as safe area is also similar for entropy $(0.401 \pm 0.048)$ and T-IMSPE $(0.394 \pm 0.039)$.

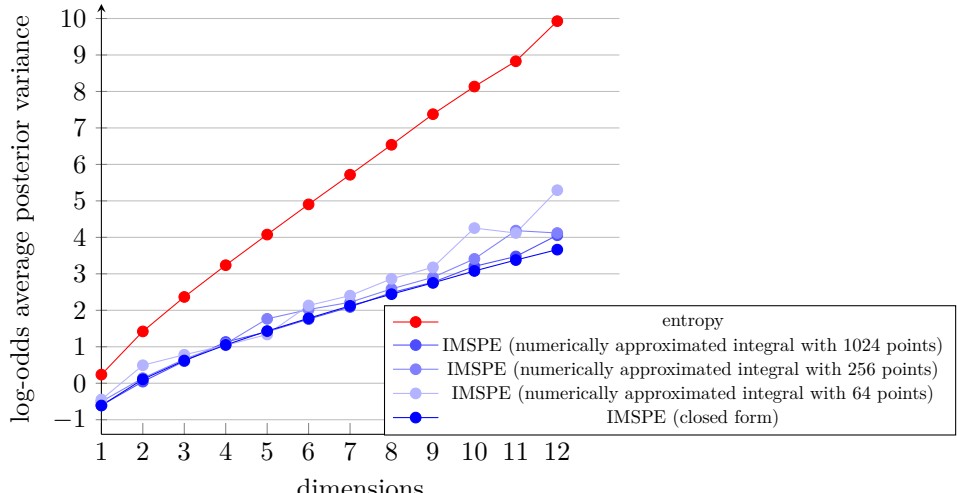

Figure 7: For various dimensions $1 \leq d \leq 12$, we consider the average posterior variance over the interval $[-1, 1]^d$ of a GP after adding a two datapoints. The first datapoints is at the origin, and the second datapoint it optimized with 5 different acquisition functions. For a better visual representation, we print the average variance $\sigma_{avg}^2$, which is bounded between 0 and 1, in log-odds, i.e. as $\log\left(\frac{\sigma_{avg}^2}{1-\sigma_{avg}^2}\right)$, where smaller is still better. The prior GP is has a squared exponential covariance functions with all hyperparameters set to one. The entropy acquisition functions (red) is very suboptimal, since the second point is always placed at a position with maximal variance after adding the origin, which is the boundary of the domain. While the numerical approximations to the intergral in the definition of IMSPE (shades of light blue) come close to the symbolically computed IMSPE (dark blue), they rarely find the same optimum. For example in one dimension, the optimally places points are at $-1$ (entropy), $-0.633$ (IMSPE approximated with 64 points), $-0.598$ (IMSPE approximated with 256 points), $-0.536$ (IMSPE approximated with 1024 points), which are at best near the real optimum at $-0.532$ (closed form IMSPE).

For the example about seasonal change from Subsection 5.1 we conducted an ablation study about the effect on dwindling superiority of T-IMSPE over entropy, once the seasonal effect weakens. We consider learning a system with seasonal changes of various changes, again by rotating the domain in the function

$$\text{McCormick}: (x_1, x_2) \mapsto$$
$$\sin(x_1 + x_2) + (x_1 - x_2)^2 - 1.5x_1 + 2.5x_2 + 1,$$

by the formula

$$\text{ssnl}_a : (t, x_1, x_2) \mapsto -1 + \frac{1}{10}\text{McCormick}\Big($$
$$\frac{1}{2}\cos\left(\frac{1}{2}\sin\left(\frac{at}{10}\right)\right)x_1 - \frac{1}{2}\sin\left(\frac{1}{2}\sin\left(\frac{at}{10}\right)\right)x_2,$$
$$\frac{1}{2}\sin\left(\frac{1}{2}\sin\left(\frac{at}{10}\right)\right)x_1 + \frac{1}{2}\cos\left(\frac{1}{2}\sin\left(\frac{at}{10}\right)\right)x_2\Big),$$

where $a \in \{0, 1, 2, 3, 4, 5\}$ determines the strength (or speed) of the seasonal change. Note that in Subsection 5.1 we used $a = 5$. The value $a = 0$ corresponds to no seasonal change. We keep all other parameters as in the main experiments.

In Figure 8 we see the results of this experiment. Entropy is superior for $a \in \{0, 1\}$, whereas for $a \geq 2$ we have that T-IMSPE results in lower RMSE values. This is in accord with both the literature as discussed in Subsection 2.4 and the results in this paper: entropy is the superior acquisition function when temporal changes are non-existant or weak, whereas T-IMSPE is the superior acquisition function for applications with temporal changes.

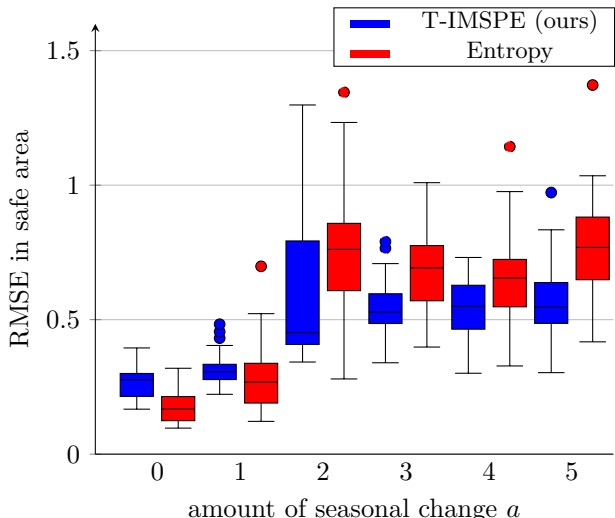

Figure 8: Box plots of the RMSE values in the safe area of the experiments from active learning with seasonal changes in Appendix E.1. We compare the results of 25 runs between T-IMSPE (blue) and entropy (red) on for various values of the strength of the seasonal change $a$. We see that T-IMSPE is superior to entropy for larger values of $a$, whereas for smaller values of $a$ we see the superiority of the entropy.

It remains an open practical question how to detect, whether a system is sufficiently enough time changing, where we currently have no clear answer. Computation time is comparable between T-IMSPE, entropy and IMSPE, so this is cannot guide a decision. We checked the length scales in direction of time as criterion, and they were not sufficiently correlated with time variance. Perhaps one could use model selection for Gaussian processes: if a GP that is constant in time direction is more suitable to model the data, then the system might not be sufficiently time variant; this is speculative. It remains to guess for a potential user of T-IMSPE, entropy, or IMSPE, which acquisition function is more suitable in a situation. If a system is noticably time variant, we suggest T-IMSPE. Note that IMSPE behaves exactly as T-IMSPE if a system is constant in time; hence, it might be a safe choice to always use (T-)IMSPE. In the borderline of mild or no time variance, all methods show comparable performance and our preliminary experiments point to a strong dependence on the dataset; in this overlap, the choice is not too relevant.

### E.2 Further results for the drift experiment

Consider the experiment about drift in examples as discussed in Subsection 5.2. Due to the drift, any active learning scheme won't bring down the RMSE significantly. Instead, the goal is to hold the RMSE on an acceptable level despite the drift. In the main paper, we have shown the RMSE gain of T-IMSPE over entropy in Figure 4. Figure 9 also shows the raw RMSE values.

### E.3 Further results for the rail pressure experiment

Subsection 5.3 considered the rail pressure experiment for dynamic systems. Here, we show additional results. Also, the amount of safe area that is recognized as safe area is drastically larger for T-IMSPE, see Figure 10. Table 2 shows some additional numerical results. In addition, this tables serves as an ablation for the maximal distance in an elipsiodal norm, allowed by the safe active learning algorithm in this experiment. This axis-parallel ellipse with semi-axes 80.3 and 2.17 is actually a circle in scaled space (see Appendix F.1 for the formula for scaling) of radius 0.1. Increasing the radius to 0.15 or 0.2, i.e. allowing more dynamic behaviour, massively decreases the quality of the results in safe active learning.

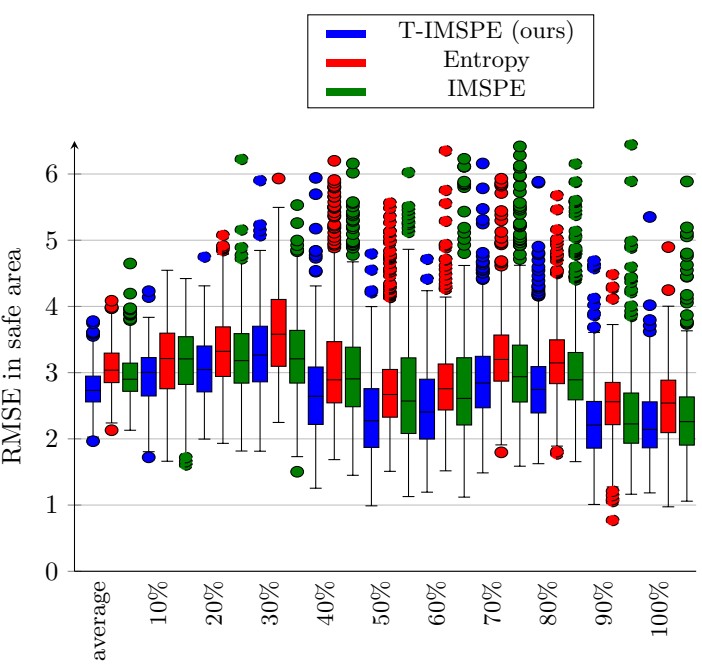

Figure 9: Box plots of the RMSE values in the safe area of the experiments from active learning under drift in Subsection 5.2. We compare the results of 500 runs between T-IMSPE (blue), entropy (red), and IMSPE (green) on average during the runs (left) and then ascending at specific time steps. In contrast to the other use cases, the drift prevents a major decrease of the RMSE over time. These are the raw values of Figure 4 from the main paper, where the difference of experiments with the same seeds were considered.

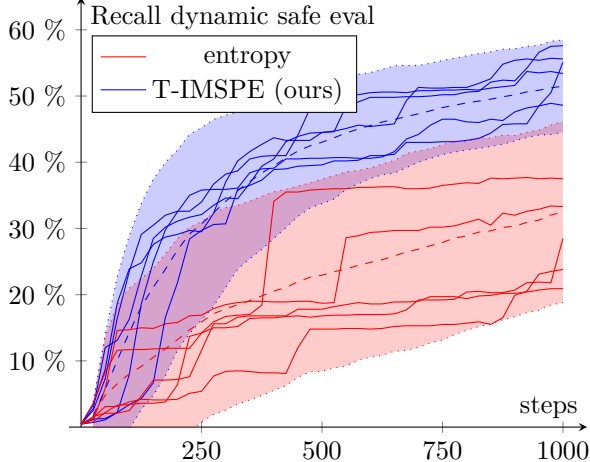

Figure 10: This diagram show the increase in recall for classifying the safe validation data correctly into as safe in the rail pressure model from Subsection 5.3. Entropy (red) show a slow incline, whereas our approach T-IMSPE rises faster, resulting in much higher recalls. The dashed line is the mean of 100 runs, the area shows the $2\sigma$ area and the solid lines show the same 5 exemplary runs as Figure 6.

## F   Technical Details on experiments

Our implementation is done in the PyTorch environment (Paszke et al., 2017).

All our GPs use the squared exponential covariance function with a separate length scale $\ell_i$ as hyperparameter for each input (automatic relevance determinantion). Additionally, we use the signal variance $\sigma_f^2$ and noise

Table 2: We show various additional results on dynamic railpressure experiment from Subsection 5.3. We compare entropy (first three rows) to T-IMSPE (seconod three rows) with three different maximal distances (0.1, 0.15, and 0.2) for each step in scaled space. The experiment in the paper used the clear superior distance of 0.1. The first row shows the computation time in seconds on an NVIDIA RTX3080 GPU of all 1000 safe active learning steps. Here, entropy clearly takes longer, even though in theory it should have a fast evaluation time by a small constant factor. Debug outputs indicates that entropy leads to significantly more failed optimization runs, where the safety and other constraints could not be kept. Both approaches choose over 90 % of points as being safe, with T-IMSPE achieving 99 % safety. Our safety criterion of two standard deviations of the GP model prediction stay below the safety bound would predict that—ignoring model error—independently chosen points are about 97.7 % safe. The last row shows the recall of our safety criterion on safe trajectories, i.e. the percentage of safe trajectories in the test data, which are recognized as being safe. In these recall values, T-IMSPE is more than twice as good as entropy. See also Figure 10, where the recall values are plotted over time, instead of the average as here. Results are the mean of all five seeds, with standard deviation in brackets.

| strategy | max. distance | comp. time (s) | safe points (%) | static recall (%) |
|----------|:-------------:|:--------------:|:---------------:|:-----------------:|
| | 0.1 | 13093($\pm$334) | 97.5($\pm$0.4) | 18.3($\pm$6.4) |
| Entropy | 0.15 | 14022($\pm$653) | 94.0($\pm$0.3) | 16.5($\pm$2.6) |
| | 0.2 | 15604($\pm$1129) | 90.3($\pm$0.2) | 9.2($\pm$2.3) |
| | 0.1 | **3792**($\pm$212) | **99.0**($\pm$0.4) | **38.5**($\pm$3.4) |
| T-IMSPE (ours) | 0.15 | 3915($\pm$179 | 97.1($\pm$0.7) | 32.8($\pm$3.6) |
| | 0.2 | 4305($\pm$271) | 93.3($\pm$1.2) | 25.6($\pm$7.2) |

variance $\sigma_n^2$ as as additional hyperparameters, as in (Rasmussen & Williams, 2006). The GP has a constant mean function $m$ as hyperparameter. For the initial training of the GP hyperparameters, we use the SQP[10] implementation from PyGranso (Liang et al., 2022; Curtis et al., 2017). After each new measurement, we retrain all hyperparameters with 30 steps of ADAM. When training hyperparameters, instead of minimizing the negative log likelihood, we minimizing the negative log a-posteriori. This is the same GP for both acquisition functions entropy and T-IMSPE.

The choice of a GP with squared exponential covariance function for all experiments might be unintiutive. Specific covariance functions might have improved the results of the examples, e.g. periodic or cosine covariance functions for the experiment with seasonal chance in Section 5.1. However, there are several reasons for experiments with a standard covariance function such as the squared exponential covariance.

1. We did not want to assume any specific time-varying structure (other than dynamic structure in the third experiment) in our covariance function.

2. We prefer a consistent approach over several experiments.

3. The GP used in the seasonal change experiment in Section 5.1 does not directly model periodic behavior. Despite this, T-IMSPE still steers the data acquisition such that even a GP with a uninformative covariance functions learns the periodic behavior, see Figure 4. We see it as an advantage to T-IMSPE that periodic structures are automatically captured with a standard Gaussian process.

4. We conducted preliminary experiments with a periodic covariance function in the seasonal change experiment in Section 5.1. These experiments used the numerical approximations to T-IMSPE, since there is no closed form version for the periodic covariance in T-IMSPE, see Table 1. The results were as follows: Safe active learning behaved very suboptimal with *all acquisition functions* when we trained the hyperparameter for the period length of the periodic covariance function, since the hyperparameter for the period length was hard to learn in a Gaussian process with multiple inputs and few measurement points. This lead to many local optima, most of which showed very suboptimal

---

[10]Without constraints, this is basically a Newton optimization.

behavior, and very overfit models. When we manually set the period to the correct value (something that seems to be unreasonable in many practical examples) and used a numerical approximation to T-IMSPE, entropy was superior to the numerically approximated T-IMPSE. This is consistent (and one of many reasons) for our observation in Appendix D that numerically evaluated integrals perform suboptimally.

We choose the following priors for the experiments. For the seasonal change experiment from Subsection 5.1 and for the drift experiments from Subsection 5.2 the priors are

$$\text{softplus}^{-1}(\ell_t) \sim \mathcal{N}(5, 1^2)$$
$$\text{softplus}^{-1}(\ell_x) \sim \mathcal{N}(0, 1^2)$$
$$\text{softplus}^{-1}(\sigma_f) \sim \mathcal{N}(1, 1^2)$$
$$\text{softplus}^{-1}(\sigma_n) \sim \mathcal{N}(-3, 1^2)$$
$$m \sim \mathcal{N}(10, 0.01^2)$$

for $\text{softplus}(x) = \log(1 + \exp(x))$, temporal length scale $\ell_t$ and spatial length scale $\ell_x$. For the rail pressure experiments from Subsection 5.3 the priors are

$$\text{softplus}^{-1}(\ell) \sim \mathcal{N}(0.5, 0.1^2)$$
$$\text{softplus}^{-1}(\sigma_f) \sim \mathcal{N}(0.5, 0.1^2)$$
$$\text{softplus}^{-1}(\sigma_n) \sim \mathcal{N}(-3, 0.1^2)$$
$$m \sim \mathcal{N}(11.77, 0.01^2).$$

For the rail pressure example, it is particularly important to have conservative priors, since otherwise safety constraints are kept much less often. The priors for the mean are particularly strict. This prevents small mean functions and hence exploration into unsafe area which are deemed safe due to unsuitable extrapolation. Taking the mean as trainable hyperparameter with prior instead of fixing it allows for flexible models once enough data is collected. While using GPyTorch (Gardner et al., 2018), we reimplemented the priors to avoid inconsistencies in GPyTorch.

The optimization for safe active learning is done again using the SQP implementation from PyGranso with 3 random restarts. In case of the first or second unsuccessful optimization, where no point keeping the constraints could be found, we start an additional optimizations with new starting points, such that at most 5 restarts are performed. In the optimizer, all tolerances are set to $10^{-4}$ and maximal 200 iterations are allowed. The most important constraints, the safety constraint, is taken $m_{\text{safe}}(x_*|x, y) + 2\sqrt{k_{\text{safe}}(x_*|x)} < c$, where $c$ is the constraint and $m_{\text{safe}}(x_*|x, y)$ resp. $\sqrt{k_{\text{safe}}(x_*|x)}$ are the mean resp. standard deviation of the posterior GP at $x_*$. This corresponds to $\alpha = 0.977$.

We adapted entropy (mildly) to the time-variant case: we maximize the variance by only allowing the current time step, similar to the usage of the acquisition function in (Fiducioso et al., 2019). However, to the best of the authors' knowledge, IMSPE is the only acquisition function that allows to specify that knowledge should be collected to add information for future time steps.

The finite measure for T-IMSPE in the experiment with seasonal changes and for the drift experiments are chosen as $\mathbf{1}_{D \times [t_0, t_0+10]}$, where $t_0$ is the current time point, $D$ is the allowed spatial domain, and $\mathbf{1}$ is the indicator function. In other words, we want to reduce the variance equally over $D$ and up to 10 time steps into the future. For the rail pressure experiment, we chose $\mathbf{1}_{D^4}$ as measure for T-IMSPE, to search for trajectories that reduce the variance over the space of all trajectories. The exponent 4 in $D^4$ is due to having 4 time steps in our NX model.

As in the seasonal change example, the drift example starts with 8 initial measurements at times $0, \ldots, 7$ positioned at the inital points of a Sobol sequence in the safe area. Afterwards, 100 further measurements at times $8, \ldots, 107$ are conducted according to the two respective safe active learning criteria entropy and T-IMSPE.

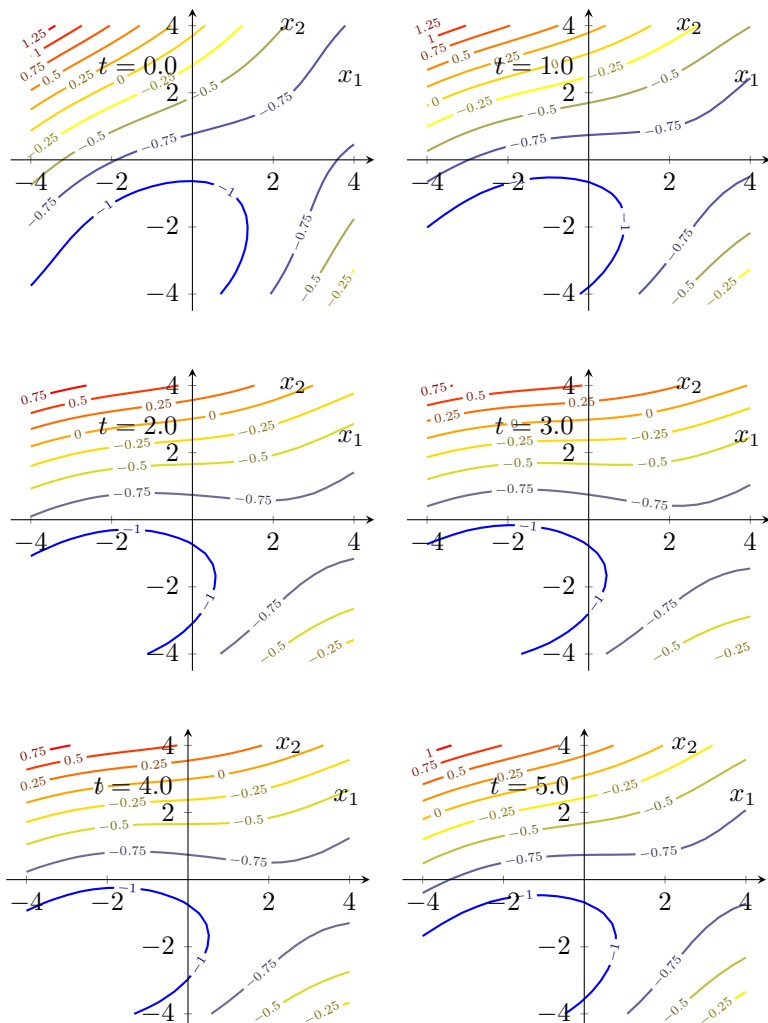

Figure 11: Plots of the function from (7) for the seasonal change experiment for various values of $t$.

In the rail pressure example, we also model real world safety procedure, mirroring a similar procedure in (Zimmer et al., 2018; Tebbe et al., 2024). Once a measurement is not safe, we jump back to the area that was initially declared as safe. Note that the jump back itself is not necessarily safe itself, due to potentially being a long trajectory. However, once inside the area that was initially declared as safe, the behavior will quickly stabilize, which is not necessarily the case outside of the initially safe domain. The NX structure was chosen similar to (Zimmer et al., 2018; Tebbe et al., 2024), i.e. based on expert knowledge. If, in a different application, this expert knowledge is not available, one can use techniques for NX structure optimization to find a suitable value, e.g. from (Yassin et al., 2010). The domain is $D = [1000, 4000] \times [0, 60]$, the safety constraint is to keep the rail pressure below 18, and the initially known safe domain for steady state behavior is $[2093, 2414] \times [14.36, 23.0]$. Bigger jumps result in uncontrollable dynamic behavior, hence we restrict $(n_{k+1}, v_{k+1})$ to an ellipse around $(n_k, v_k)$ with axis-parallel semi-axes 80.3 and 2.17. We choose 256 initial measurements in the known safe domain, while keeping distances between points short enough that the initial measurements are all safe. After an unsafe measurements with rail pressure above 18, we will return to the safe domain to stabilize the dynamic behavior.

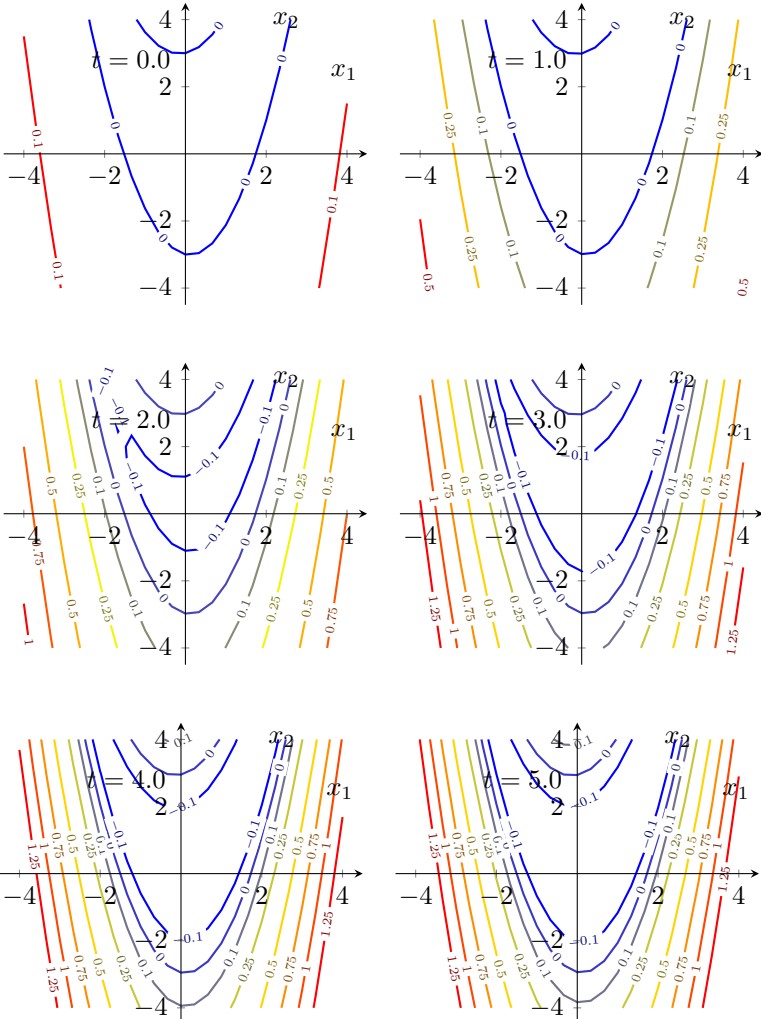

Figure 12: Plots of the function from (8) for the drift experiment for various values of $t$.

## F.1 Formulas for the experiments

The formula of the seasonal change experiment in Subsection 5.1 uses the classical McCormick function (McCormick, 1976)

$$\text{McCormick} : (x_1, x_2) \mapsto$$
$$\sin(x_1 + x_2) + (x_1 - x_2)^2 - 1.5x_1 + 2.5x_2 + 1,$$

in the formula

$$\text{ssnl} : (t, x_1, x_2) \mapsto -1 + \frac{1}{10} \text{McCormick}\Big($$
$$\frac{1}{2}\cos\left(\frac{1}{2}\sin\left(\frac{t}{2}\right)\right)x_1 - \frac{1}{2}\sin\left(\frac{1}{2}\sin\left(\frac{t}{2}\right)\right)x_2, \tag{7}$$
$$\frac{1}{2}\sin\left(\frac{1}{2}\sin\left(\frac{t}{2}\right)\right)x_1 + \frac{1}{2}\cos\left(\frac{1}{2}\sin\left(\frac{t}{2}\right)\right)x_2\Big).$$

See Figure 11 for a visual representation of this function. We consider this function in the spatial domain $[-4, 4]^2$, where the area $[-0.5, 0.5] \times [-1, 1]$ is deemed save initially. Measurements at $(t, x_1, x_2)$ are deemed safe if $\text{ssnl}(t, x_1, x_2) < 0$.

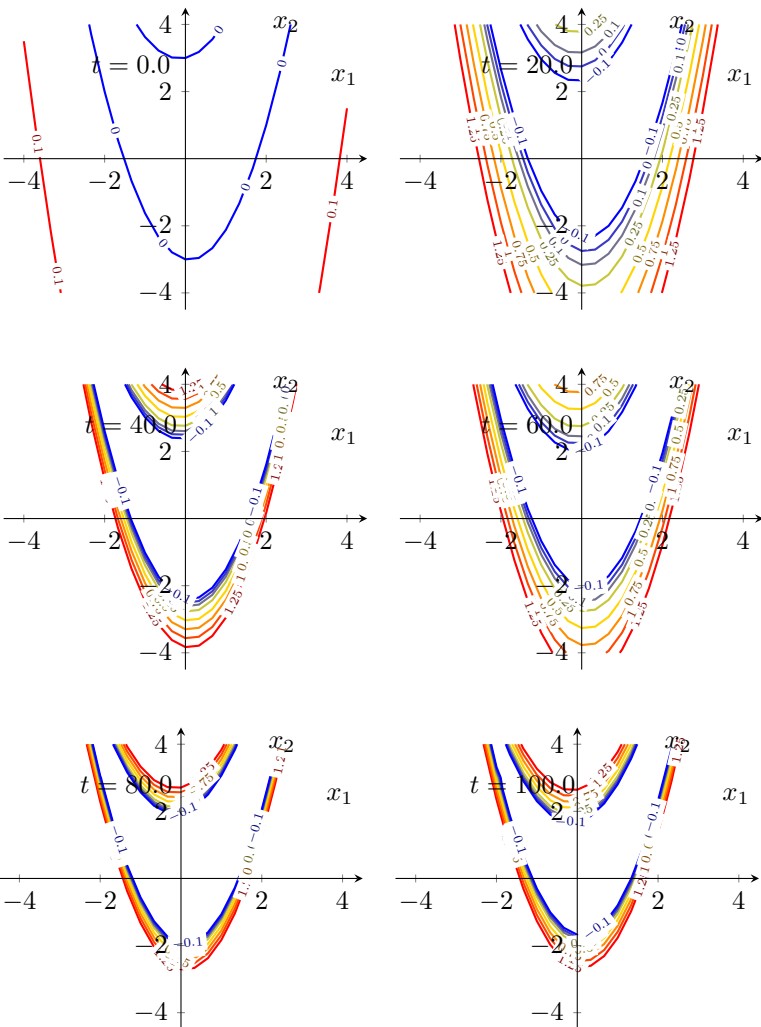

Figure 13: Plots of the function from (8) for the drift experiment for various values of $t$.

The formula of the drift experiment in Subsection 5.2 is given by

$$
\begin{aligned}
\mathrm{drft} : (t, x_1, x_2) \mapsto \frac{1}{1000} & \left( \left( 2 + \sin \left( \frac{t}{2} \right) \right) t + 1 \right) \\
& \cdot \left( \mathrm{Rosenbrock}(x_1, x_2) - 25 + \frac{t}{10} \right)
\end{aligned}
\tag{8}
$$

for the classical Rosenbrock function (Rosenbrock, 1960)

$$
\mathrm{Rosenbrock} : (x_1, x_2) \mapsto (8 \cdot |x_1^2 - x_2| + (1 - x_1)^2).
$$

This function is visualized in Figure 12 and Figure 13. We consider this function in the spatial domain $[-4, 4]^2$, where the area $[-0.5, 0.5] \times [-1, 1]$ is deemed save initially. Measurements at $(t, x_1, x_2)$ are deemed safe if $\mathrm{drft}(t, x_1, x_2) < 0$. Note that the safe area is decreasing as $t$ increases.

