# OpenReview forum: "Future-aware Safe Active Learning of Time Varying Systems using Gaussian Processes"
_TMLR — Accepted by TMLR_

### Review · Reviewer_3k2z · 2024-11-25

**Summary Of Contributions:**

The submission introduces T-IMSPE, a new acquisition function for safe active learning in time-varying systems that minimizes posterior variance over both current and future states. The authors show the closed-form computability of T-IMSPE, and thus its $O(n^3)$ and $O(n^2)$ time-complexity in training and testing, respectively. Finally, empirical results demonstrate that T-IMSPE outperforms the traditional method of entropy as a measure for information in modeling quality while adhering to safety constraints.

**Audience:**

Yes

**Claims And Evidence:**

Yes

**Requested Changes:**

1. It would be interesting to investigate the performance (both time and MSE) of the T-IMSPE method with sparse GPs (variational inference) compared with the exact closed-form computation of the posterior. Sparse GPs should allow for drastically improving the computational complexity.
2. From what I understood, T-IMSPE is the objective that is used in the safe active learning and instead of $I(\cdot)$ in equation (1), correct? I think it would be better if this is explicitly mentioned, because in the current version of the paper, equation (1) and the pseudocode are never referenced after Section 2.2.
3. The work assumes correct model specification. What happens if the correct model (e.g., kernel parameters or even kernel type) is not given? How well does the method perform compared with the entropy method?

### Minor comments
1. Typo on page 3: "This **yield** the probability..." should be "This **yields** the probability..."
2. Typo on page 5: "In both cases, T-IMSPE **work**..." should be "In both cases, T-IMSPE **works**.

**Strengths And Weaknesses:**

### Strengths
1. The paper is well written and easily understood.
2. Adding a time dimension to IMSPE is done intuitively, also allowing easy closed-form computation.
3. The experiment section is nicely done (kudos on having p-values), and clearly shows better performance compared with entropy and IMSPE.

### Weaknesses & questions
1. The T-IMSPE is not applicable to all kernels. This is all the more reason to investigate sparse GPs/variational inference, where an inexact posterior is computable and a closed-form is not necessary.
2. From what I understood, tuning the kernel parameters and computing the posterior (i.e., training) is done first, and then the T-IMSPE method is to be used, correct? This is why it is claimed that "T-IMSPE \[has\] the same computational complexity as computing the entropy."
3. I am a bit confused as to why the authors restricted test data to the safe area. They mention that "model quality is only of interest there," but wouldn't it also be important to see if the methods are sampling "unsafe" points?
4. In Section 5 you reference Appendix F and "the attached code," but I did not see attached/linked code anywhere.

---

> ### Author Response · Authors · 2025-02-26
> **Response to Reviewer 3k2z**
>
> We thank you for taking the time to write a constructive and very helpful review! It helped us to clearly improve the paper.
>
> > The T-IMSPE is not applicable to all kernels. This is all the more reason to investigate sparse GPs/variational inference, where an inexact posterior is computable and a closed-form is not necessary. It would be interesting to investigate the performance (both time and MSE) of the T-IMSPE method with sparse GPs (variational inference) compared with the exact closed-form computation of the posterior. Sparse GPs should allow for drastically improving the computational complexity.
>
> Sparse GPs (in the sense of Titsias'09 or Hensman et al.'13) are also GPs with standard kernels, but having the amount of data reduced. In that sense, sparse GPs are absolutely compatible with T-IMSPE. There is obvious some overlap between sparse GPs and active learning in the literature. However, sparse GPs are usually used for a lot of data, whereas active learning is used in applications of scarse data. We see such tests as orthogonal to our paper, as our appendices are already rather long.
>
> > From what I understood, tuning the kernel parameters and computing the posterior (i.e., training) is done first, and then the T-IMSPE method is to be used, correct? This is why it is claimed that "T-IMSPE [has] the same computational complexity as computing the entropy."
>
> Yes, your understanding is correct. We clarified this point in the paper before Corollary 4.5 by writing "We follow the state of the art approach in active learning, where GPs are trained after each addition of a new data point, including hyperparameter training. After this training, IMSPE and T-IMSPE have the same computational complexity as computing the entropy and can be evaluted on $O(n^2)$ for $n$ training data points."
>
> > I am a bit confused as to why the authors restricted test data to the safe area. They mention that "model quality is only of interest there," but wouldn't it also be important to see if the methods are sampling "unsafe" points?
>
> We have clarified this point, see also W10 and the comments regarding Remark 2.1 by Reviewer RkUu.
> You are correct in that only the safe region is relevant for the model quality. We have added a more detailed discussion in Remark 2.1: "In active learning, points are not drawn independently for several reasons. First, the Gaussian process might be misspecified in general. Second, safe active learning approaches predominately collect safe data, which leads Gaussian processes to have extrapolating predictions that are considered safe. Third, the model misspecification might be specifically large, as points evaluated in active learning might be far from previous measurements. Due to this lack of statistical independence, points might have a higher probability of being unsafe."
> That being said, all experiments give numbers in the text, how many points are unsafe.
>
> > In Section 5 you reference Appendix F and "the attached code," but I did not see attached/linked code anywhere.
>
> You are correct. We forgot to add the code to the supplementary material. It is now added.
>
> > From what I understood, T-IMSPE is the objective that is used in the safe active learning and instead of  in equation (1), correct? I think it would be better if this is explicitly mentioned, because in the current version of the paper, equation (1) and the pseudocode are never referenced after Section 2.2.
>
> Good point, we have added a sentence after Equation (3) (the new number in the revision of the former Equation (1).)
>
> > The work assumes correct model specification. What happens if the correct model (e.g., kernel parameters or even kernel type) is not given? How well does the method perform compared with the entropy method?
>
> You are correct in that model misspecification is a potential problem. As in the start ort the art, we use the SE covariance function that yields to realizations that are dense in many function spaces, such as the set of continuous or smooth functions. Hence, complete model misspecifications is impossible. However, in the view of scarse data, the question of more suitable priors and hyperparameters is relevant. Here, the cookbook provided in our paper allows to construct suitable kernels for many different behaviors. The hyperparameters are rather unstable early, but stabilize quickly. Hence, additional unsafe points might appear early on.

---

> ### Comment · Reviewer_3k2z · 2025-03-03
> **Reply to authors' rebuttal**
>
> I thank the authors for their comments and revisions. They have adequately addressed my concerns and questions, and I have no further comments.

---

> > ### Author Response · Authors · 2025-03-06
> > **Reply to reply to authors' rebuttal**
> >
> > We thank the reviewer for their time and constructive feedback in making our paper better!

---

### Review · Reviewer_EHv6 · 2024-12-05

**Summary Of Contributions:**

This paper introduces T-IMSPE (Time-aware Integrated Mean Squared Prediction Error), a novel acquisition function designed for safe active learning in time-varying systems. The method aims to minimize posterior variance across both current and future states, addressing the challenges posed by system dynamics such as drift and seasonal changes.

**Audience:**

Yes

**Broader Impact Concerns:**

No concern

**Claims And Evidence:**

Yes

**Requested Changes:**

- more reference and connections with safe RL with Gaussian Process in recent literature should be discussed
- more intuitive explanation, comments should be provided to the theoretical proof and theories themselves
- safe robotics benchmark experiments are required

**Strengths And Weaknesses:**

**Strength**:
- This paper extends the IMSPE acquisition function to account for future states, enabling better data collection in systems with temporal dependencies.
- I like that the paper provides guidelines on which GP kernels, domains, and measures are suitable for T-IMSPE, and explores its theoretical boundaries.
- The proofs are technically solid
- Interesting toy problems and real-world scenario experiments are evaluated, which demonstrates the applicability of T-IMSPE.

**Weakness**:
- the connections and differentiations of safe learning methods with SAL are poorly explained. Most safe learning tries to acquire dynamics model data safely, and the purpose is fundamentally different the formulation presented in this paper. Additionally, there are more literature on the safe reinforcement learning with Gaussian Process in recent years, the author should comment their connections with those works.
- Though the theoretical proofs are sufficient, the author dives deeply into the mathematical aspects of T-IMSPE without providing sufficient intuition or simplified examples to guide non-specialist readers. This approach may alienate practitioners and those outside the immediate research niche, limiting the paper’s broader impact.
- The experiments are interesting but the impact is limited. I don't think those are well established benchmarks. The authors should at least validate the applicability of T-IMSPE in a broader set of safe exploration tasks, such as safe reinforcement learning benchmark tasks. This will answer the scalability to high dimensional robotics tasks. Good platforms include:
     - Safety Gymnasium: A Unified Safe Reinforcement Learning Benchmark
     - Guard: A Safe Reinforcement Learning Benchmark
- Figures and diagrams, while useful, are dense and lack adequate annotation for clarity. For example, Figure 1 requires significant prior knowledge to interpret, which limits its instructional value.

---

> ### Author Response · Authors · 2025-02-26
> **Response to Reviewer EHv6**
>
> We thank you for taking the time to write a constructive and helpful review! It helped us to improve the paper.
>
> >  the connections and differentiations of safe learning methods with SAL are poorly explained. Most safe learning tries to acquire dynamics model data safely, and the purpose is fundamentally different the formulation presented in this paper. Additionally, there are more literature on the safe reinforcement learning with Gaussian Process in recent years, the author should comment their connections with those works.
>
> Thank you for your feedback. Our paper focuses on active learning, which is distinct from reinforcement learning. While we acknowledge the growing body of literature on safe reinforcement learning with Gaussian Processes, our work does not fall within this scope. Instead, our primary concern is the safe acquisition of informative data for model learning, rather than decision-making policies under uncertainty.
>
> In our original submission, we already cited 100 relevant works, carefully selecting those most pertinent to our focus. We even largely excluded Bayesian optimization, despite its closer relationship to active learning compared to reinforcement learning. Expanding our discussion to include safe reinforcement learning would, in our view, dilute the clarity and focus of our contribution. However, if the reviewer has specific references that provide particularly relevant insights or highlight key conceptual overlaps, we would be happy to consider discussing them in a revised version.
>
> > Though the theoretical proofs are sufficient, the author dives deeply into the mathematical aspects of T-IMSPE without providing sufficient intuition or simplified examples to guide non-specialist readers. This approach may alienate practitioners and those outside the immediate research niche, limiting the paper’s broader impact.
>
> Thank you for your suggestions. Yes, the motivation of IMSPE and its comparison against entropy was buried in the appendices. Hence, we moved parts of Examples A.1 , inclusing formerly Figure 4, now Figure 2, and Appendix D.2 into Subsection 2.3. Example A.1 gives an intuition for IMSPE on a single datapoint and Appendix D.2 discussed the differences between IMSPE and the state of the art entropy.
>
> > The experiments are interesting but the impact is limited. I don't think those are well established benchmarks. The authors should at least validate the applicability of T-IMSPE in a broader set of safe exploration tasks, such as safe reinforcement learning benchmark tasks. This will answer the scalability to high dimensional robotics tasks. Good platforms include:
> >  -Safety Gymnasium: A Unified Safe Reinforcement Learning Benchmark
> >  -Guard: A Safe Reinforcement Learning Benchmark
> > safe robotics benchmark experiments are required
>
> We appreciate the reviewer’s interest in broader validation. However, the benchmarks suggested—such as Safety Gymnasium and Guard—are specifically designed for safe reinforcement learning (RL), where the primary focus is on learning safe policies for sequential decision-making. In contrast, our work is centered on active learning (AL), where the goal is the efficient and informative acquisition of data for model learning rather than policy optimization.
>
> Safe RL benchmarks typically involve high-dimensional control tasks with continuous interaction and reward-based learning, which differs fundamentally from the selective, query-based nature of AL. Applying T-IMSPE to such benchmarks would conflate objectives and obscure the core contributions of our method. Instead, we have designed our experiments to rigorously evaluate safe data acquisition under constraints relevant to AL.
>
> > Figures and diagrams, while useful, are dense and lack adequate annotation for clarity. For example, Figure 1 requires significant prior knowledge to interpret, which limits its instructional value.
>
> Thank you for your feedback. We acknowledge that Figure 1 requires prior knowledge to fully interpret. When designing it, we carefully considered how to balance completeness with clarity. While the figure may have limited instructional value at first glance, we believe it becomes much more informative after reading the first two sections of the paper, where the necessary context is provided.
>
> That said, we recognize the importance of accessibility. We tries to explore ways to improve annotation and clarity to enhance its interpretability, even for readers encountering it for the first time. If the reviewer has specific suggestions for elements that require further clarification, we would be happy to incorporate them.

---

### Review · Reviewer_RkUu · 2025-02-07

**Summary Of Contributions:**

The manuscript proposes an active learning framework that aims at both data efficiency and safety during learning. A system is considered that is time-varying, which includes relevant aspects in practice like drifts and seasonal changes. The T-IMSPE criterion is proposed for active sampling, which minimizes posterior variance over a future horizon. The proposed criterion and corresponding active learning method is compared to established methods in numerical experiments, as well as an example with real-world data.

**Audience:**

Yes

**Broader Impact Concerns:**

No concerns.

**Claims And Evidence:**

No

**Requested Changes:**

Please address and/or respond to the above-mentioned weaknesses.

**Strengths And Weaknesses:**

# Strengths

(S1) The problem considered in this manuscript---active learning for data efficiency *and* safety---is highly relevant and the suggested approach seems to carry some originality.

(S2) The authors show that the proposed integrals can be computed in closed form, which is of practical benefit.

(S3) An interesting aspect of the paper is the notion of "elementary functions that can be computed in closed form in PyTorch", etc. (Sec. 4.3).  This appears to be very relevant.  I am wondering though whether such arguments and formalizations have been considered in other areas of computer science like automatic differentiation, etc.  There seems to be no discussion of related work on these aspects; it might be worth adding.

(S4) Numerical experiments illustrate the superior quantitative performance over state of the art criteria for active sampling.


# Weaknesses

(W1) My main difficulty is that I have trouble pin-pointing the exact technical problem that is being solved. Essentially, I am missing a precise problem formulation.  It seems that Sec. 2 somehow introduces the problem that is being considered, without spelling this out explicitly.  This would help for clarification and transparency of what is actually the technical problem that is considered in this work. Furthermore, Sec. 4, first line talks about "time-varying models" without making these precise.  Sec. 4.1 seems to combine a problem statement (what is the time-dependent model/problem) with a solution. It would probably be clearer if this was disentangled.

I would like to ask the authors to consider adding a specific (sub)-section at an appropriate place early in the paper, where the technical system is formalized (including the "type" of system, cf. (W2), and the time dependence that is considered) and where the problem that shall be solved in this work is precisely stated. Owing to the lack of a precise problem formulation, I have difficulty identifying the core technical problem that was solved in this work.



(W2) *One* core aspect of (W1) is that it is not clear to me what type of system the authors are considering. Is it a dynamical system, x_t+1 = f(x_t, ...) (i.e., one with a state/memory), or a static system, y = f(x)? Note: both can be time varying (i.e., have explicit dependence of f on time t).

* This should be stated as part of a precise problem formulation (W1)
* What system is considered has important implications on how sampling can be done. For example, in the Introduction, I was puzzled by the sentence "where measurements can be taken at almost arbitrary positions in a continuous domain" At first, after reading title and abstract, I was assuming that the authors would do active learning for dynamical systems, where samples (usually state/action combinations) can *not* be chosen freely, as the system has to "evolve" to reach a certain state x_t (see, e.g., Buisson-Fenet et al 2020).  If they do consider static systems, it might be possible to freely choose the input.

* I suggest to discuss the conceptual difference where sampling in dynamical systems is different as one cannot freely choose the input, but has to "go through the physics/dynamics" instead.  A potential place where the authors could elaborate on this might be Sec. 3.2 (related work).  At least, this discussion should reflect the different nature of the sampling and contrast other works with what is done in this work.



(W3) The list of contributions should be improved:

* "We provide several theoretical contribution" is too vague for a list of contributions; it should be spelled out (or at least an indication given) what results are obtained.
* In the list of contributions, I would have found it insightful to read what "real world examples" are being considered.
* It is not clear to me how "We distinctly delineate the theoretical scope of closed-form integrals" is related to the discussion so far in the introduction; hence, the contributions appears out of context (at this stage).



(W4) Safe active learning (SAL) setting (Sec. 2.2): It would be helpful to clarify a bit better how the proposed setting is different from the state of the art.  For example, training of safe GPs (g_safe = GP(...)) is not new AFAIK, so references should be added.  Then, is the formulation (1) a new one, or also existing work?  It did not become clear to me what is new and what is background here, partly due to missing references.  It would be good if this was clarified.

Furthermore, is the considered SAL setting one where there exists a TRUE safety function g_safe, but we don't know it (which would be more of a frequentist setting), or is there no such function (more Bayesian).  In particular, I was wondering about (1), whether the safety constraints refers to the true one, or to our information about it given the current data.  This I would expect to have consequences on the algorithm and the type of guarantees that one may be able to give. In my opinion, it would be critical to clarify the setting.  Both settings exist in the literature, but they can have consequences regarding the type of bounds, guarantees, etc. that are available (see, e.g., [R1, R2])



(W5) The role of hyperparameter optimization is not clear in the context of the proposed SAL framework. On p. 4, "GP hyperparaemter training is well established and fast", this sentence appears a bit out of context.  Furthermore, hyperparameter optimization can have a significant effect on safety, at least in other safe learning settings (see e.g. [R2]), so it would be good to discuss 1) if it is also the case here, and 2) if hyperparameter optimization is used at all (and of what type).



(W6) Sec. 2.3: IMO, this is too short as this seems to be a main basis for what is proposed herein.  First, reference(s) should be given.  Second, I would appreciate understanding the key idea as to why IMSPE is chosen, and preferred to entropy, for example.  Details can be deferred to the appendix (as is currently the case), but the main ideas should be in the main paper.



(W7) Related work: The paper states that it considers active learning for *time-varying* systems.  There is recent work on BO with time-varying objectives, which is not reviewed in related work, however.  I was wondering whether this is relevant.  In particular, there is also recent work on safety *and* time-varying problems [R3, R4], which should probably be discussed in this context. Orthogonal to this, also see (S3) regarding related work.



(W8) The technical developments in Sec. 4 require clarification.

* It should be discussed what parts of the model in (3) are novel, and which parts are known.  For example, using GPs that have time as an argument is rather common in BO with time variations / spatio-temporal GPs.
* I did not understand the idea of Sec. 4.2.  Does (4) still include the time dependency?  It seems to be removed, but this method is still titled T-IMSPE (with the "T" in the acronym).
* Sec. 4.2, first/second line: is y_t = f(x_t, ...) on purpose?  I thought this was the dynamics with x_t+1 = f(x_t, ...)
* Sec. 4.2, line 4:  What does the notation $x_{i,t}$ (double indices) mean?  Has this been introduced?



(W9) In Sec. 4.4, the authors present a "cookbook" for GP kernels and those fulfilling Def. 4.3, however referring to the appendix for all the details.  As this is an interesting aspect and result of this work, it might be worth to present the kernels / results that compose the cookbook in 1-2 tables in the main paper, while leaving the details in the appendix.  This could also include the references to where (some of) the results have been known in prior work.



(W10) Presentation and discussion of experimental results.

* The description of the experiment in Sec. 5.1 is insufficient. From the description, one cannot understand what the system is about, why it is challenging / relevant in the context of this work, etc.  The main paper should be self-contained without the need to go to the supplementary material for main aspects. Just referencing an equation in the appendix is not a good structure IMHO and appears like these results were just pushed to the appendix without reconsidering the structure (maybe to make the page limit...).  Similarly for Sec. 5.2.
* Experiments in Sec. 5.1:  It seems that the evaluation (test data) is restricted to the safe area only.  I understand that model quality should be evaluated there.  However, isn't it also of interest how often samples in the unsafe areas were taken and how this compares between the methods?  IMO, this could (and maybe should) also be discussed and maybe shown with appropriate plots (not just one number in the text).
* The results (e.g., Figure 2) illustrate that the proposed criterion (T-IMSPE) is quantitatively superior to the other criteria.  However, it wasn't so clear how relevant this improvement is from the application point of view.  Based on the presentation, I get the impression that the improvement is there, but I'm not so sure that it is super relevant. I wonder if some discussion / clarification can be added in this regard.



(W11) In parts, I found the writing and polishing level of this manuscript to be below standard. Examples:

* Figure 1 does not seem to be referenced from anywhere in the text.
* For the pseudo-code (p. 4), I suggest to use a proper algorithm environment.
* What exactly is/are the algorithm(s) that is being run in the experiments?  Please make this explicit, e.g., referencing to pseudo code + equation reference to the criterion.
* At times, I got the impression that components of the work were moved to the appendix, without taking much care of how this affects the flow and understanding in the main paper.  At least for me, I found the number of references to the appendix a bit disturbing and would have liked the main paper to be more self-contained (e.g., by making sure that all main statements/results/insights are in the main paper). As an example, in the intro of Sec. 5, the main thoughts behind the experimental setups should be briefly discussed in the main paper.  Another example, see comments (W9) + (W10).
* The paper has a super extensive appendix. While this *might* be needed for the results in the paper, I recommend to make an effort to re-evaluate if the appendix can be shortened.
* Bottom of page 8, is there a duplication of text?  The last two paragraphs appear very similar...
* Please do a careful pass for typos, such as punctuation.  *Examples*:
  * Page 1, forth line from the bottom -> period instead of semicolon?
  * end of Sec. 2.1 missing a period.


# Minor comments and suggestions

* Introduction: "Many systems pose challenges for experimentation due to their high costs, whether in actual implementation or simulation," -- I'm not quite sure if I can follow the authors argumentation.  They argue that experiments on hardware or simulation are often expensive; then they say that data-driven simulation is better.  So, simulation replaces simulation in a sense.  I believe the authors are getting at surrogate models, etc., but I suggest to 1) clarify what is meant here, 2) consider if this is the intended motivation here (because there might be other motivations as well for looking into active learning for dynamical systems).
* "DoE is not even applicable in the face of safety constraints" -- I don't understand this statement.  I would consider Bayesian Optimization (BO) a method for DoE.  And Safe BO is a kind of DoE, where one exactly considers safety constraints when deciding on the next configuration.  Hence, this statement needs to be clarified or modified, and it seems that Safe BO should be more properly discussed here.
* Please explain what NX structures mean (in the Intro or in Sec. 2.1).
* p. 2, "we start from the acquisition function IMSPE ..." -> References should be added for the different acquisitions functions, IMO.
* p. 3, Remark 2.1: "Therefore, these points are not statistically independent random draws of a Gaussian" -- I don't understand this sentence.  In particular, why does the non-iid property follow from the previous sentence (points being far away...)
* p. 3, Remark 2.1: What does "slightly lower" mean quantitatively for being "acceptable"?  The statement is not technically precise.
* Check writing "Pytorch" vs. "PyTorch" for consistency



# References (of this review)

[R1] Fiedler et al., Practical and rigorous uncertainty bounds for gaussian process regression, AAAI 2021

[R2] Fiedler et al., On Safety in Safe Bayesian Optimization, TMLR 2024

[R3] Jialin Li, et al., Safe Time-Varying Optimization based on Gaussian Processes with Spatio-Temporal Kernel, NeurIPS 2024

[R4] Holzapfel et al., Event-triggered safe Bayesian optimization on quadcopters, L4DC 2024

---

> ### Author Response · Authors · 2025-02-26
> **Response to Reviewer RkUu, part 1**
>
> We thank you for taking the time to write a very extensive, constructive and very helpful review! It helped us to clearly improve the paper.
>
> > (S3) An interesting aspect of the paper is the notion of "elementary functions that can be computed in closed form in PyTorch", etc. (Sec. 4.3). This appears to be very relevant. I am wondering though whether such arguments and formalizations have been considered in other areas of computer science like automatic differentiation, etc. There seems to be no discussion of related work on these aspects; it might be worth adding.
>
> Thank you for your interest! We are not aware of other applications of the theory of elementary functions in machine learning and related fields as automatic differentiation. This comes with low certainty, since the search term "elementary function" appears in unrelated papers rather often.
>
> > (W1) My main difficulty is that I have trouble pin-pointing the exact technical problem that is being solved. Essentially, I am missing a precise problem formulation. It seems that Sec. 2 somehow introduces the problem that is being considered, without spelling this out explicitly. This would help for clarification and transparency of what is actually the technical problem that is considered in this work. Furthermore, Sec. 4, first line talks about "time-varying models" without making these precise. Sec. 4.1 seems to combine a problem statement (what is the time-dependent model/problem) with a solution. It would probably be clearer if this was disentangled.
> > I would like to ask the authors to consider adding a specific (sub)-section at an appropriate place early in the paper, where the technical system is formalized (including the "type" of system, cf. (W2), and the time dependence that is considered) and where the problem that shall be solved in this work is precisely stated. Owing to the lack of a precise problem formulation, I have difficulty identifying the core technical problem that was solved in this work.
> > (W2) One core aspect of (W1) is that it is not clear to me what type of system the authors are considering. Is it a dynamical system, x_t+1 = f(x_t, ...) (i.e., one with a state/memory), or a static system, y = f(x)? Note: both can be time varying (i.e., have explicit dependence of f on time t).
> > This should be stated as part of a precise problem formulation (W1)
>
> We have added and additional Subsection 2.2 "Time varying systems", which adresses this point and several points in your review below.
>
> This section describes the relevant systems we are considering with the background where such systems appear. The short version is that we are both considering systems of the form (1) $y=f(x,t)$ and (2) $y=f(x_t,x_{t-1},x_{t-2},\ldots,x_{t-\ell+1})$.
>
> > What system is considered has important implications on how sampling can be done. For example, in the Introduction, I was puzzled by the sentence "where measurements can be taken at almost arbitrary positions in a continuous domain" At first, after reading title and abstract, I was assuming that the authors would do active learning for dynamical systems, where samples (usually state/action combinations) can not be chosen freely, as the system has to "evolve" to reach a certain state x_t (see, e.g., Buisson-Fenet et al 2020). If they do consider static systems, it might be possible to freely choose the input.
> > I suggest to discuss the conceptual difference where sampling in dynamical systems is different as one cannot freely choose the input, but has to "go through the physics/dynamics" instead. A potential place where the authors could elaborate on this might be Sec. 3.2 (related work). At least, this discussion should reflect the different nature of the sampling and contrast other works with what is done in this work.
>
> We have added an explanation to the new Subsection 2.2, clarifying this point:
>
> In contrast, in dynamical systems, the combinations of state and action cannot be chosen freely, as the system has to evolve to reach a certain state $x_t$. In our time varying systems, we have complete freedom to choose parts of the inputs, i.e.\ $x$ in (1) respectively $x_t$ in (2), and the influence of the time comes from unconstrollable conditions respectively a memory of the system that influces the system output.

---

> ### Author Response · Authors · 2025-02-26
> **Response to Reviewer RkUu, part 2**
>
> > (W3) The list of contributions should be improved:
> > "We provide several theoretical contribution" is too vague for a list of contributions; it should be spelled out (or at least an indication given) what results are obtained.
> > In the list of contributions, I would have found it insightful to read what "real world examples" are being considered.
> > It is not clear to me how "We distinctly delineate the theoretical scope of closed-form integrals" is related to the discussion so far in the introduction; hence, the contributions appears out of context (at this stage).
>
> We have rephrased the vague and out of context contribution as "(T-)IMSPE involves the computation of an integral. We consider computability of (T-)IMSPE in closed form via antiderivatives and provide a cookbook of usage for various Gaussian processes kernels and domains to be used in (T-)IMSPE, see Section 4.". We also specified the examples in the second contribution as "We demonstrate the advantages of T-IMSPE in terms of model quality in a real world example about engine calibration for dynamic driving and two toy examples about drift and seasonal changes, see Section 5.".
>
> > (W4) Safe active learning (SAL) setting (Sec. 2.2): It would be helpful to clarify a bit better how the proposed setting is different from the state of the art. For example, training of safe GPs (g_safe = GP(...)) is not new AFAIK, so references should be added. Then, is the formulation (1) a new one, or also existing work? It did not become clear to me what is new and what is background here, partly due to missing references. It would be good if this was clarified.
> > Furthermore, is the considered SAL setting one where there exists a TRUE safety function g_safe, but we don't know it (which would be more of a frequentist setting), or is there no such function (more Bayesian). In particular, I was wondering about (1), whether the safety constraints refers to the true one, or to our information about it given the current data. This I would expect to have consequences on the algorithm and the type of guarantees that one may be able to give. In my opinion, it would be critical to clarify the setting. Both settings exist in the literature, but they can have consequences regarding the type of bounds, guarantees, etc. that are available (see, e.g., [R1, R2])
>
> In the "Background" section, we have added references to the the beginning of the "Safe active learning" subsection to make clear that the safety constraints we are considering is well-known in the literature. In this subsection, we have also clarified the assumptions about the safety function that we are following: "We consider the case of a safety critical quantity, e.g. temperature should stay below a critical temperature. Then, one can calculate a safety indicator $z$ that indicates safety for non-negative values $z\ge0$. We assume that the safety critical quantity cannot be computed, but (potentially noisy) measurements on this safety critical quantity can be obtained."
>
> > (W5) The role of hyperparameter optimization is not clear in the context of the proposed SAL framework. On p. 4, "GP hyperparaemter training is well established and fast", this sentence appears a bit out of context. Furthermore, hyperparameter optimization can have a significant effect on safety, at least in other safe learning settings (see e.g. [R2]), so it would be good to discuss 1) if it is also the case here, and 2) if hyperparameter optimization is used at all (and of what type).
>
> You are correct that hyperparameter optimiztaion is a critical point, which we have clarified as follows:
> "We use GP hyperparameter training, which is fast and well-established. If in certain real-time applications further speed up is required, techniques of amortized inference are at hand (Bitzer et al., 2023; Liu et al., 2020). Changing hyperparameters has a strong effect on the overall performance of all algorithms, including on safety (Fiedler et al., 2024) due to model misspecification, in particular in case of few data points. Still, changing hyperparameters is state of the art in active learning and Bayesian optimization (Garnett, 2023) and, in our experience, superior to fixing hyperparameters."

---

> ### Author Response · Authors · 2025-02-26
> **Response to Reviewer RkUu, part 3**
>
> > (W6) Sec. 2.3: IMO, this is too short as this seems to be a main basis for what is proposed herein. First, reference(s) should be given. Second, I would appreciate understanding the key idea as to why IMSPE is chosen, and preferred to entropy, for example. Details can be deferred to the appendix (as is currently the case), but the main ideas should be in the main paper.
>
> Thank you for your suggestions. Yes, the motivation of IMSPE and its comparison against entropy was buried in the appendices. Hence, we moved parts of Examples A.1 , inclusing formerly Figure 4, now Figure 2, and Appendix D.2 into Subsection 2.3. (Example A.1 gives an intuition for IMSPE on a single datapoint and Appendix D.2 discussed the differences between IMSPE and the state of the art entropy.)
>
> > (W7) Related work: The paper states that it considers active learning for time-varying systems. There is recent work on BO with time-varying objectives, which is not reviewed in related work, however. I was wondering whether this is relevant. In particular, there is also recent work on safety and time-varying problems [R3, R4], which should probably be discussed in this context. Orthogonal to this, also see (S3) regarding related work.
>
> Thank you for bringing these references up. We intentionally decided not to give an overview of related BO approaches, as more than 100 references seemed to be enough. For example in the related work section of [R3] alone, additional 27 papers on safe Bayesian optimization are being cited, which all could/should be mentioned here. Hence, we did not include these references.
>
> > (W8) The technical developments in Sec. 4 require clarification.
> >  It should be discussed what parts of the model in (3) are novel, and which parts are known. For example, using GPs that have time as an argument is rather common in BO with time variations / spatio-temporal GPs.
>
> Thx, we added references in the new Subsection 2.2, describing our setup. Obviously, these model approaches mentioned there are very well-known in the literature.
>
> >  I did not understand the idea of Sec. 4.2. Does (4) still include the time dependency? It seems to be removed, but this method is still titled T-IMSPE (with the "T" in the acronym).
> >  Sec. 4.2, first/second line: is y_t = f(x_t, ...) on purpose? I thought this was the dynamics with x_t+1 = f(x_t, ...)
>
> The new Subsection 2.2 describes the setups and different forms of time dependency. These are referenced in Subsection 4.2 (and Subsection 4.1). This also describes the setup for $y_t = f(x_t, x_{t-1},...)$. There, the time is not an input of the model, but the NX-structure of the models takes previous input into consideration as a memory of the past.
>
> > (W9) In Sec. 4.4, the authors present a "cookbook" for GP kernels and those fulfilling Def. 4.3, however referring to the appendix for all the details. As this is an interesting aspect and result of this work, it might be worth to present the kernels / results that compose the cookbook in 1-2 tables in the main paper, while leaving the details in the appendix. This could also include the references to where (some of) the results have been known in prior work.
>
> Thx for your interest! We are not aware on prior work on this anywhere in machine learning.
> We would gladly move parts of the appendices A.2 oder A.3 up into the main paper. However, the main paper makes due with the definition of $P$-elementary covariance marginalizable functions, whereas the appendix uses the generalization of $P$-elementary cross-covariance marginalizable functions. After some discussion, we have the feeling that moving this up would require additional motivation and in the end basically result in moving up all 4 pages of appendices A.2 and A.3 into the main paper. Hence, we are hesitant to do this.
> All major results are summarized in Subsection 4.4.
>
> > (W10) Presentation and discussion of experimental results.
> > The description of the experiment in Sec. 5.1 is insufficient. From the description, one cannot understand what the system is about, why it is challenging / relevant in the context of this work, etc. The main paper should be self-contained without the need to go to the supplementary material for main aspects. Just referencing an equation in the appendix is not a good structure IMHO and appears like these results were just pushed to the appendix without reconsidering the structure (maybe to make the page limit...). Similarly for Sec. 5.2.
>
> Thx for catching this. We added a description of the difficulty of the experiments in the main paper for all three examples.

---

> ### Author Response · Authors · 2025-02-26
> **Response to Reviewer RkUu, part 4**
>
> > Experiments in Sec. 5.1: It seems that the evaluation (test data) is restricted to the safe area only. I understand that model quality should be evaluated there. However, isn't it also of interest how often samples in the unsafe areas were taken and how this compares between the methods? IMO, this could (and maybe should) also be discussed and maybe shown with appropriate plots (not just one number in the text).
>
> There is no difference in the safety approach between different acquisition functions. Hence, we would not expect and the number we presented in the paper did not show significant differences in the numbers of safe points. Due to little data and resulting model misspecification, slightly more unsafe measurement points are taken in the initial phases of the active learning algorithm. This is to be expected. Otherwise, we did not spot any interesting patterns.
>
> > The results (e.g., Figure 2) illustrate that the proposed criterion (T-IMSPE) is quantitatively superior to the other criteria. However, it wasn't so clear how relevant this improvement is from the application point of view. Based on the presentation, I get the impression that the improvement is there, but I'm not so sure that it is super relevant. I wonder if some discussion / clarification can be added in this regard.
>
> You are correct that the many repetitions ensured that effects are clearly visible.
>
> We have now added that the percentage of reduction in quality (which is perhaps around 10%-20% between entropy and T-IMSPE, with IMPSE being somewhere in the middle) is only one criteria. Another criterion is the measurement time needed to reach a certain model error. Here, in the drift example, the median error reached by T-IMSPE at 30% of the time is reached by entropy at 50% of the time, which would result in a 40% reduction of measuremt time and cost. Similarly, the mean error reached by T-IMSE in the rail pressure example after 250 steps is the same as the one reached by entropy after 1000 steps. This is an 75% reduction of measurement time and cost.
>
> We have stressed this point in the figures of the paper.
>
> > (W11) In parts, I found the writing and polishing level of this manuscript to be below standard. Examples:
> Figure 1 does not seem to be referenced from anywhere in the text.
>
> Done.
>
> > For the pseudo-code (p. 4), I suggest to use a proper algorithm environment.
>
> We have changed the environment from the package "algpseudocode" with default settings to the environment of "algorithm2e" with default settings and adapted the algorithm into a floating environment.
>
> > What exactly is/are the algorithm(s) that is being run in the experiments? Please make this explicit, e.g., referencing to pseudo code + equation reference to the criterion.
>
> Done.
>
> > At times, I got the impression that components of the work were moved to the appendix, without taking much care of how this affects the flow and understanding in the main paper. At least for me, I found the number of references to the appendix a bit disturbing and would have liked the main paper to be more self-contained (e.g., by making sure that all main statements/results/insights are in the main paper). As an example, in the intro of Sec. 5, the main thoughts behind the experimental setups should be briefly discussed in the main paper. Another example, see comments (W9) + (W10).
>
> We added some texts making the paper more self-contained. We think that references to additional experiments, ablations, and details in the appendix are helpfull, even though many of them are given in the paper.
>
> > The paper has a super extensive appendix. While this might be needed for the results in the paper, I recommend to make an effort to re-evaluate if the appendix can be shortened.
>
> We have shortened the appendix by 7 papes by removing the rather long equation for the rail pressure experiments. They are given in the code (which we have added now).
>
> > Bottom of page 8, is there a duplication of text? The last two paragraphs appear very similar...
>
> Thanks, one of them is removed.

---

> ### Author Response · Authors · 2025-02-26
> **Response to Reviewer RkUu, part 5**
>
> Minor comments and suggestions
>
> > Introduction: "Many systems pose challenges for experimentation due to their high costs, whether in actual implementation or simulation," -- I'm not quite sure if I can follow the authors argumentation. They argue that experiments on hardware or simulation are often expensive; then they say that data-driven simulation is better. So, simulation replaces simulation in a sense. I believe the authors are getting at surrogate models, etc., but I suggest to 1) clarify what is meant here, 2) consider if this is the intended motivation here (because there might be other motivations as well for looking into active learning for dynamical systems).
>
> We have improved the wording.
>
> > "DoE is not even applicable in the face of safety constraints" -- I don't understand this statement. I would consider Bayesian Optimization (BO) a method for DoE. And Safe BO is a kind of DoE, where one exactly considers safety constraints when deciding on the next configuration. Hence, this statement needs to be clarified or modified, and it seems that Safe BO should be more properly discussed here.
>
> We clarified in the paper that we meant static DoEs.
>
> > Please explain what NX structures mean (in the Intro or in Sec. 2.1).
>
> This is now explained in the new Subsection 2.2.
>
> > p. 2, "we start from the acquisition function IMSPE ..." -> References should be added for the different acquisitions functions, IMO.
>
> We have taken the most seminal of the IMSPE references into the introduction.
>
> > p. 3, Remark 2.1: "Therefore, these points are not statistically independent random draws of a Gaussian" -- I don't understand this sentence. In particular, why does the non-iid property follow from the previous sentence (points being far away...)
> > p. 3, Remark 2.1: What does "slightly lower" mean quantitatively for being "acceptable"? The statement is not technically precise.
>
>
> We have clarified this part: "In active learning, points are not drawn independently, since the data points are not drawn independently but chosen by an acquisition function. Due to this lack of statistical independence, points might have a higher probability of being unsafe."
>
> We have also specified that we would consider 95 % safety as being acceptable.

---

> > ### Comment · Reviewer_RkUu · 2025-03-03
> > **Response to authors' response**
> >
> > The authors have responded to most of my comments and provided a revised version of their manuscript.  In my opinion, the paper has improved considerably.  Thank you for the revision!  I am glad that you found most of my comments helpful.
> >
> > There are still some aspects, where I think the manuscript can be further improved.  Examples of what I still consider somewhat unclear (but I leave it up to the authors to decide):
> >
> > 1) I appreciate that the authors now explicitly mention that similar problems have been addressed with BO ("Optimizing such systems (1) or (2) with Bayesian optimization is common, see e. g. (Brunzema et al., 2022)."). I also understand that the authors do not want to cite various BO references.  However, after the above sentence, IMHO it should be better clarified, what is the difference of this work to BO then.  Certain BO variants also use aquisition functions for safe active learning in time-varying systems (see previous comments).  So is this work proposing an alternative to existing works, but there is no conceptual difference to BO?; and the novel aquisition function could, for example, also be used in BO?
> >
> > 2) I still think it would be helpful to clarify the conceptual difference to sampling in dynamical systems in Sec. 3.2, e.g., by referring to Buisson-Fenet et al 2020.
> >
> > Finally, I think that the paper can use some further polishing in some parts and with regards to my previous comments, but I leave it up to the authors how far they want to go.

---

> > > ### Author Response · Authors · 2025-03-06
> > > **Response to response to authors' response**
> > >
> > > Thank you for the additional feedback and for acknowledging the improvements in our revision. We appreciate your constructive comments.
> > >
> > > For the camera-ready version, we intend to make the following changes:
> > > - We will clarify the distinction between our approach and Bayesian Optimization (BO): AL aims at learning the whole input space e.g. in order to build a digital twin, perform some post-hoc or worst-case analysis while BO is only interested in an optimal point of operation and, once this point or the area is found, the rest of the input space is not of any interest anymore. Therefore, despite some technical similarities, AL and BO are conceptually very different. Specifically, our acquisition function, which reduces the variance globally, cannot be integrated meaningfully into a BO framework, as far as we know.
> > > - We will explicitly address the conceptual differences between our approach and sampling in dynamical systems in Section 3.2 by incorporating a reference to Buisson-Fenet et al. (2020).
> > > - We will read our manuscript for the camera-ready version and refine it to further improve clarity and readability.

---

### Author Response · Authors · 2025-02-26
**Response to the reviews.**

We thank all reviewers for their comments which, in out opinion, have resulted in significant improvements in the article (and have increased its length). In the revised version, changes are marked.

The most important issues, raised by both RkUu and 3k2z, was a clear desciption of time varying system. To address this we have added a new Subsection 2.2, which motivates and describes the time varying system used in our paper. The remainder of our response is contained in the comments to each individual reviewer.

---

### Decision · Action_Editor_6fRH · 2025-04-03

**Recommendation:** Accept as is

**Comment:**

See above.

**Audience:**

Yes. There is an audience for this paper in the safe active learning community when time-varying systems are involved.

**Claims And Evidence:**

Claims are well supported and supported by convincing evidence. The authors have made a great effort to place their work in the right context.